# Non-canonical reader modules of BAZ1A promote recovery from DNA damage

Mariano Oppikofer [1,7], Meredith Sagolla[2], Benjamin Haley[3], Hui-Min Zhang[4], Sarah K. Kummerfeld [5], Jawahar Sudhamsu[6], E. Megan Flynn[1], Tianyi Bai[6,8], Jennifer Zhang[4], Claudio Ciferri[6] & Andrea G. Cochran [1]

Members of the ISWI family of chromatin remodelers mobilize nucleosomes to control DNA accessibility and, in some cases, are required for recovery from DNA damage. However, it remains poorly understood how the non-catalytic ISWI subunits BAZ1A and BAZ1B might contact chromatin to direct the ATPase SMARCA5. Here, we find that the plant home-odomain of BAZ1A, but not that of BAZ1B, has the unusual function of binding DNA. Furthermore, the BAZ1A bromodomain has a non-canonical gatekeeper residue and binds relatively weakly to acetylated histone peptides. Using CRISPR-Cas9-mediated genome editing we find that BAZ1A and BAZ1B each recruit SMARCA5 to sites of damaged chromatin and promote survival. Genetic engineering of structure-designed bromodomain and plant homeodomain mutants reveals that reader modules of BAZ1A and BAZ1B, even when non-standard, are critical for DNA damage recovery in part by regulating ISWI factors loading at DNA lesions and supporting transcriptional programs required for survival.

[1] Department of Early Discovery Biochemistry, Genentech, Inc., 1 DNA Way, South San Francisco, CA 94080, USA. [2] Center for Advanced Light Microscopy, Genentech, Inc., 1 DNA Way, South San Francisco, CA 94080, USA. [3] Department of Molecular Biology, Genentech, Inc., 1 DNA Way, South San Francisco, CA 94080, USA. [4] Protein Analytical Chemistry, Genentech, Inc., 1 DNA Way, South San Francisco, CA 94080, USA. [5] Department of Bioinformatics and Computational Biology, Genentech, Inc., 1 DNA Way, South San Francisco, CA 94080, USA. [6] Department of Structural Biology, Genentech, Inc., 1 DNA Way, South San Francisco, CA 94080, USA. [7] Present address: Pfizer Worldwide Research and Development, 558 Eastern Point Rd., Groton, CT 06340, USA. [8] Present address: STEMCELL Technologies Inc., 1618 Station Street, Vancouver, BC, Canada V6A 1B6. Correspondence and requests for materials should be addressed to M.O. (email: mariano.oppikofer@pfizer.com) or to A.G.C. (email: cochran.andrea@gene.com)

I n the eukaryotic nucleus, DNA is tightly associated with histones to form chromatin fibers. The repetitive unit of chromatin is the nucleosome, where ~146 base pairs (bp) of DNA wrap around an octamer of the core histones H2A, H2B, H3, and H4[1]. While the octamer core is highly compact, ~30% of its mass is disordered in the N-terminal and C-terminal "tails", which can be highly modified post-translationally, either to open or to compact chromatin structure, and serve as docking sites for chromatin regulators[2–4].

Repair of DNA damage generally requires ATP-dependent chromatin remodelers that alter histone-DNA contacts to mobilize, evict, or replace nucleosomes[5–9]. ISWI chromatin remodelers pair one of two ATPase subunits, SMARCA5 (SNF2H) or SMARCA1 (SNF2L), with one of six regulatory subunits: BAZ1A (ACF1), BAZ1B (WSTF), BAZ2A (TIP5), BPTF, CECR2, or RSF1, and several of these factors have been implicated in DNA repair pathways[10, 11].

One such repair process, nucleotide excision repair (NER), is triggered by helix-distorting, intrastrand DNA lesions, like those generated by ultraviolet (UV) irradiation[12–14]. BAZ1A facilitates G2/M checkpoint activation after UV irradiation, ensures cell survival, and promotes transcription-coupled NER[15–17]. Moreover, GFP fusions of BAZ1A, BAZ1B, or SMARCA5 are enriched at lesions produced by UV irradiation, suggesting a direct role for ISWI complexes in NER[17].

A second major form of DNA damage is double-stranded DNA breaks (DSB) that are typically repaired by non-homologous end joining (NHEJ) or homologous recombination (HR)[18]. Experimental induction of DSB combined with RNA interference (RNAi) link the ISWI catalytic motor SMARCA5, as well as the regulatory subunits BAZ1A, BAZ1B, and RSF1, to DSB repair by both NHEJ and HR, a topic recently reviewed[11]. In particular, BAZ1B is an "atypical" protein kinase that phosphorylates Y142 of the histone variant H2A.X[19]. This in turn affects the phosphorylation of S139 (the product referred to as γ-H2AX), which is critical for recognition and repair of DSB[19, 20]. Furthermore, BAZ1A may directly recruit the Ku complex to sites of DSB to promote NHEJ[15].

Consistent with functional evidence, BAZ1A, BAZ1B, and SMARCA5 are found to be recruited rapidly to DSB[15, 16, 21–23]. SMARCA5 accumulation at sites of DNA damage is facilitated by PARP1-dependent poly-ADP-ribose chain deposition[24], the structural nuclear protein NuMA[25], and the FACT component SUPT16H[26]. In addition, the physical interaction of SMARCA5 and SIRT6 stabilizes the loading of SMARCA5 at sites of DSB[27]. The observation that SMARCA5 recruitment to DSB requires a region of SMARCA5 that interacts with BAZ1A[15] suggests that non-catalytic ISWI subunits may contribute to SMARCA5 loading as well. However, the scope of regulatory subunit involvement and detailed interactions at damage sites are only partially known.

Near their C-termini BAZ1A and BAZ1B each contain a plant homeodomain (PHD) and a bromodomain (BD). BDs are established acyl-lysine binders[28–30], while PHD domains interact with a wider range of binding partners; examples include the N-terminal region of unmodified histone H3 and methyl-lysine-containing peptides[31, 32]. However, the importance of these PHD and BD modules remains unexplored in BAZ1A and BAZ1B, either biochemically or in cellular systems.

Here, we report the engineering of BAZ1A-knockout (KO) and BAZ1B-KO human cell lines by CRISPR-Cas9-mediated genome editing. Using these lines we demonstrate that BAZ1A and BAZ1B promote recovery after DNA damage in human cells, but only in part by recruiting SMARCA5 to damaged chromatin. Moreover, we report the crystal structure of BAZ1A-BD, and we discover the ability of BAZ1A-PHD to bind DNA. Finally, using these structural and biochemical insights, we design mutants to show that the BD and PHD modules of BAZ1A, even when non-standard, each play a functional role in recovery from DNA damage.

## Results

**Loss of BAZ1A or BAZ1B impairs DNA damage recovery.** Decreasing BAZ1A expression by means of RNAi sensitizes transformed human cells and primary human fibroblasts to DNA damage[15–17, 22]. In contrast, BAZ1A-deficient mouse somatic cells extracted from Baz1a[−/−] mice appear to respond indistinguishably from wild-type cells to DNA damage[33]. This prompted us to revisit the role that BAZ1A might play in responding to DNA damage in human cells by means of genetic removal.

Using the CRISPR-Cas9 gene editing system[34–39] (reviewed in ref. [40]), we successfully abrogated BAZ1A expression in three independent clonal lines of transformed human cells (HeLa) (BAZ1A KO clones c1–c3; Fig. 1a; Supplementary Fig. 1a, b), while protein levels of the paralog ISWI regulator BAZ1B were unaffected (Fig. 1a). To assess gross effects of BAZ1A removal, we monitored cell growth with high temporal resolution by quantifying confluence (spatial coverage on a plate) at 4 h intervals. The growth rates of BAZ1A-KO cells were equal to one another and to parental cells (Fig. 1b). Next, we monitored the response to DNA damage induced by a controlled pulse of ultraviolet light (UVC, λ = 254 nm) delivered 48 h after seeding the cells. The UVC pulse causes helix-distorting base lesions such as cyclobutane pyrimidine dimers and 6–4 pyrimidine-pyrimidone photoproducts[41, 42]. In addition, UVC irradiation can cause DSBs from DNA replication stress and the production of reactive oxygen species[43–45]. The growth rate of parental cells was only mildly decreased after the UVC pulse (Fig. 1b), consistent with effective DNA damage repair. In contrast, all three BAZ1A-KO clonal cell lines showed a substantial reduction in growth rate in response to the UVC pulse (Fig. 1b). In addition to an acute pulse of DNA damage, we investigated the effect of milder (but constant) DNA damage-induced stress by monitoring cell growth in presence of a sub-lethal dose of phleomycin D1 (Fig. 1c), which binds and cleaves DNA to form both single-stranded and DSB[46]. The growth rate of parental cells was decreased only slightly in the presence of phleomycin D1. In contrast, the growth rate of BAZ1A-KO cells was significantly decreased, although for one of three clones the decrease was rather mild (Fig. 1c). Together, this indicates that genetic removal of BAZ1A impairs proliferation under various types of DNA damage-induced stress in human cells.

To investigate whether the BAZ1A paralog BAZ1B is functionally similar, we generated by CRISPR-Cas9 two independent clonal lines that lacked BAZ1B expression but that retained normal levels of BAZ1A (Fig. 1a; Supplementary Fig. 1a, b). Like BAZ1A-KO cells, BAZ1B-KO cells were hypersensitive both to a UVC pulse and to continuous phleomycin D1 treatment (Fig. 1d, e). This indicates that each of the ISWI non-catalytic subunits BAZ1A and BAZ1B plays an important role in the response of human cells to DNA damage.

One concern in genome-editing experiments is off-target cleavage by the Cas9 nuclease; however, our evaluation of independent clonal lines minimizes this concern. To further demonstrate that the lack of BAZ1A or BAZ1B was the primary cause of the DNA damage hypersensitivity shown above, we sought to rescue these phenotypes by reintroducing BAZ1A or BAZ1B to the corresponding KO cells through stable lentiviral transduction. The cDNAs of BAZ1A and BAZ1B were placed under the control of a cytomegalovirus expression cassette for constitutive expression. To assess the importance of protein levels in reconstituted cells, we cloned independent cell lines with

observable but different levels of BAZ1A expression. These include a clone with BAZ1A levels decreased compared to parental cells (low), and a clone that expresses BAZ1A at a level at least as high as that of parental cells (high) (Fig. 1f). We found that low levels of BAZ1A produced only a minor recovery, if any, in the rate of cell proliferation after DNA damage compared to *BAZ1A*-KO cells (Fig. 1g). In contrast, high levels of BAZ1A significantly rescued

the DNA damage hypersensitivity of *BAZ1A*-KO cells (Fig. 1g). This demonstrates that lack of BAZ1A is the likely cause of the DNA damage hypersensitivity of *BAZ1A*-KO cells, and that near wild-type levels of BAZ1A are required to eliminate hypersensitivity. Similarly, reintroducing BAZ1B at near wild-type levels to *BAZ1B*-KO cells (in the single clone obtained) promoted cell proliferation after DNA damage (Fig. 1h, i).

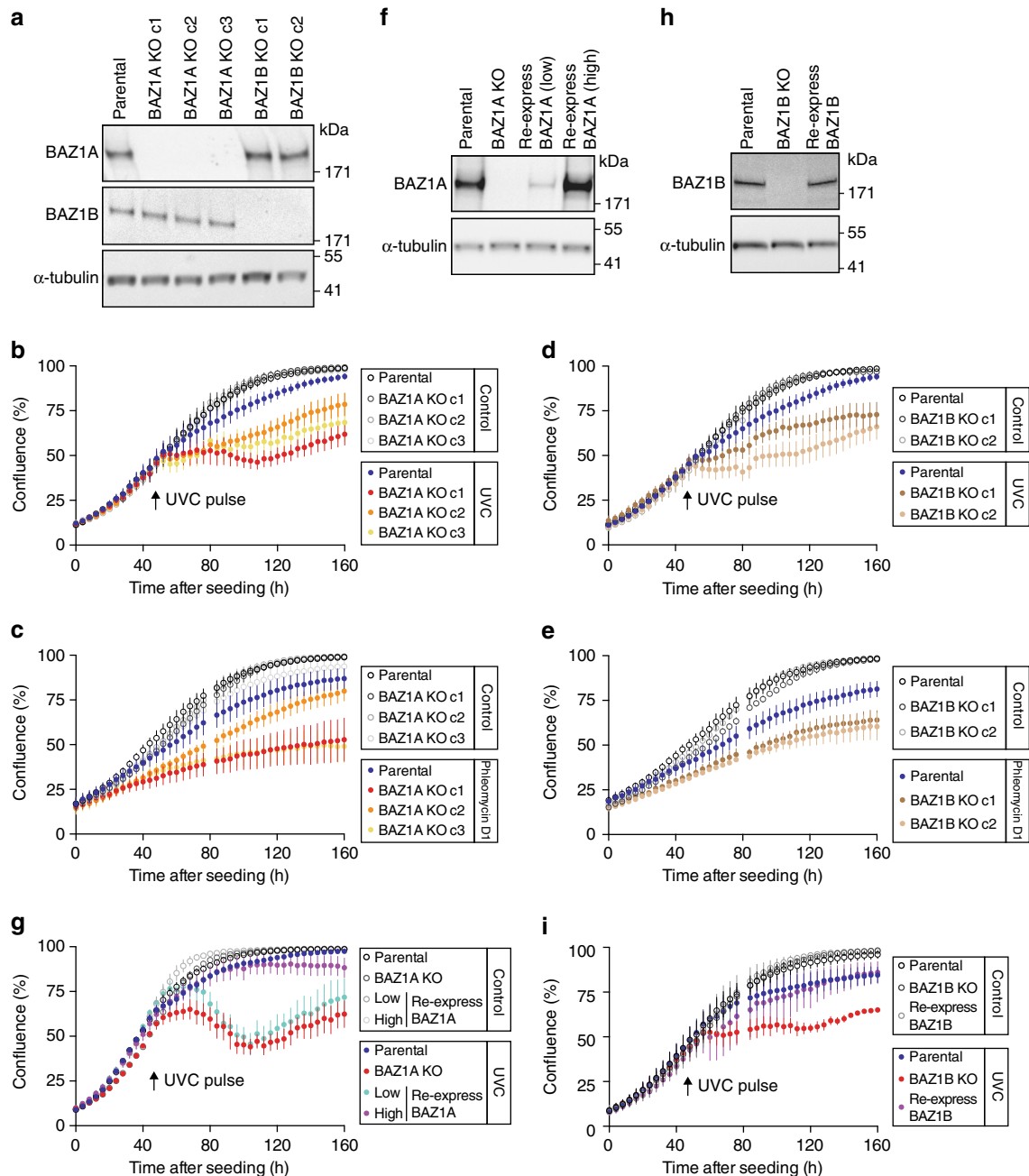

**Fig. 1** The non-catalytic ISWI subunits BAZ1A and BAZ1B promote cell growth after DNA damage. **a** Western blot analysis of BAZ1A and BAZ1B expression in parental and genome edited *BAZ1A*-KO and *BAZ1B*-KO HeLa cells. Individual clones are labeled "c1" to "c3". **b** Confluence of *BAZ1A*-KO clonal lines c1, c2, and c3 measured over time, before and after a pulse of UVC light. **c** Confluence of *BAZ1A*-KO clonal lines c1, c2, and c3 measured over time in presence of 20 µM phleomycin D1. **d** Same as **b** but for *BAZ1B*-KO clonal lines c1 and c2. **e** Same as **c** but for *BAZ1B*-KO clonal lines c1 and c2. **f** Western blot analysis of BAZ1A expression in parental, *BAZ1A*-KO (clone c1) and two independent *BAZ1A*-KO clones where BAZ1A was re-expressed at "low" and "high" levels by lentiviral integration of the cDNA of *BAZ1A*. **g** Rescue of growth defect of *BAZ1A* KO by "low" and "high" levels of BAZ1A expression. The experiment was performed as described for **b**. **h** Western blot analysis of BAZ1B expression in parental, *BAZ1B*-KO (clone c2) and a clonal line where BAZ1B was re-expressed by lentiviral integration of the cDNA of *BAZ1B*. **i** Rescue of growth defect of *BAZ1B* KO by BAZ1B re-expression. The experiment was performed as described for **b**. Each time point shown for **b**–**e**, **g** and **i** is the mean value ± s.e.m from three independent experiments

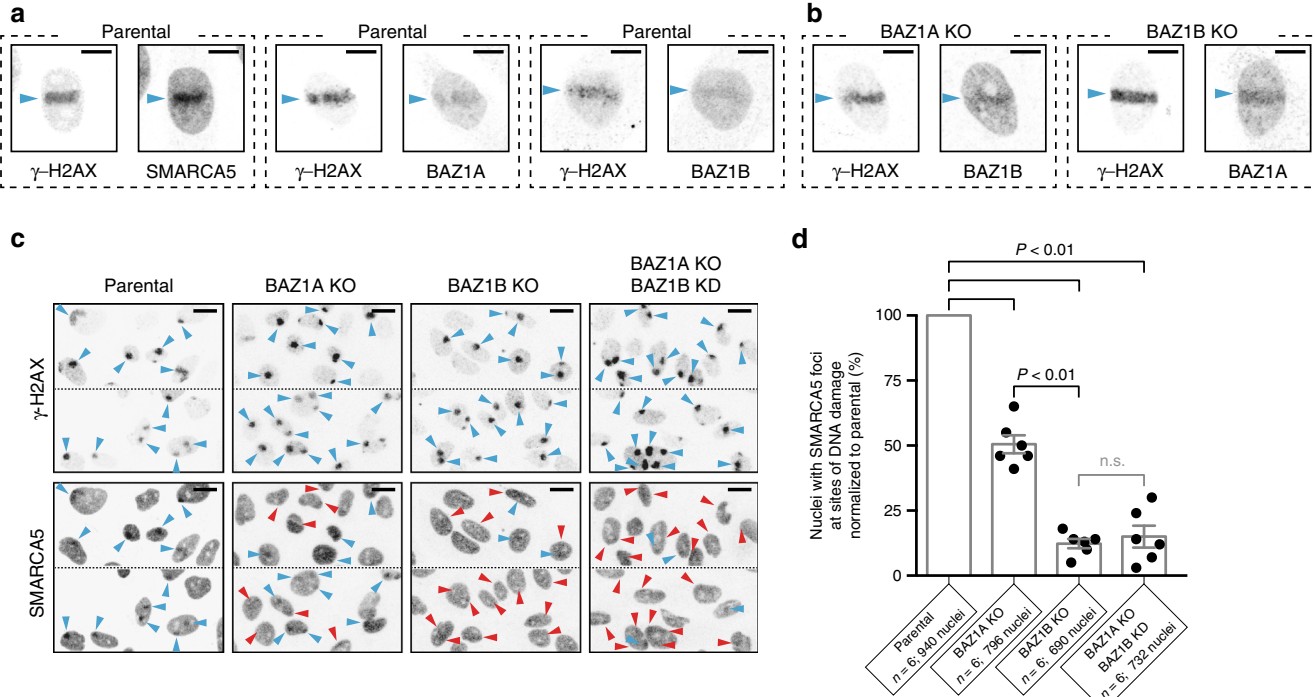

**Fig. 2** BAZ1A and BAZ1B accumulate independently to sites of DNA damage and recruit SMARCA5. **a** Accumulation of endogenous SMARCA5, BAZ1A, and BAZ1B at sites of laser micro-irradiation in parental HeLa cells. The genetic background is indicated on *top* of the panels. Regions of DNA damage are identified by the presence of γ-H2AX (histone variant H2A.X phosphorylated on S139), and are labeled with *cyan arrowheads*. Images are representative of at least five individual irradiated cells. *Scale bar*, 10 μm. **b** Accumulation of endogenous BAZ1A and BAZ1B at sites of laser micro-irradiation in genome edited *BAZ1A*-KO and *BAZ1B*-KO cells. *Scale bar*, 10 μm. **c** Accumulation of endogenous SMARCA5 at sites of DNA damage in cells subjected to UVC irradiation through a 5 μm porous membrane[47]. Each square panel is composed of two rectangular representative images showing 6–10 nuclei. The genetic background is indicated on *top* of the panels. *Cyan arrowheads* indicate sites of damage identified by the presence of γ-H2AX (*top*), with concomitant increased SMARCA5 signal (*bottom*). In contrast, *red arrowheads* indicate sites of damage without appreciable corresponding SMARCA5 signal. *Scale bar*, 20 μm. **d** Recruitment of SMARCA5 as quantified from six independent experiments is shown as histograms of the mean value ± s.e.m and normalized to recruitment in parental cells; the total number of nuclei examined for each case is reported on the figure. The values from individual experiments are shown with *black dots*. P-values ($P < 0.01$) were calculated using a two-tailed Mann–Whitney test; *n.s.* not significant ($P > 0.05$)

A previous RNAi-based study[16] indicated that loss of *BAZ1A* leads to cell death following DNA damage. Therefore, we tested whether the reduction in cell number of *BAZ1A*-KO cells after phleomycin D1-induced DNA damage was accompanied by an increase in cell death. In agreement with the earlier study, we found that *BAZ1A*-KO cells experienced elevated levels of DNA damage-induced cell death compared to parental cells (Supplementary Fig. 1c) and that BAZ1A re-expression in *BAZ1A*-KO cells promoted survival (Supplementary Fig. 1c). Together, the data demonstrate that BAZ1A plays a critical role in supporting growth after DNA damage and reducing DNA damage-induced death in a human cell line.

**BAZ1A and BAZ1B each promote SMARCA5 loading at DNA lesions**. BAZ1A, BAZ1B, and the ATPase SMARCA5 are rapidly recruited to sites of DNA damage, consistent with a critical role in repair processes[15–17, 21–23]. We validated these observations by monitoring recruitment of the endogenous proteins to sites of DNA damage induced by laser micro-irradiation (Fig. 2a; Supplementary Fig. 2a, b). Deletion of a C-terminal segment of SMARCA5 necessary for interaction with BAZ1A impairs recruitment of SMARCA5 to sites of DNA damage[15]. Therefore, we asked whether one or both of the regulatory subunits BAZ1A and BAZ1B might be necessary to recruit the ATPase SMARCA5 to chromatin lesions. First, we determined that BAZ1A and BAZ1B are recruited to sites of DNA damage independently from one another by performing laser micro-irradiation experiments in

*BAZ1A*-KO and *BAZ1B*-KO cells (Fig. 2b; Supplementary Fig. 2c). We then generated foci of DNA damage in a much larger number of cells by bromodeoxyuridine (BrdU) photosensitization coupled with UVC irradiation through 5 μm porous membranes[47]. This protocol generates DSBs, and allowed us to quantify the recruitment of SMARCA5 to DNA lesions. In parental cells a large fraction of nuclei showed focal enrichment of SMARCA5 to sites of DNA damage (Fig. 2c). In contrast, *BAZ1A*-KO cells showed a ~50% reduction in the number of nuclei with visible enrichment of SMARCA5 at sites positive for γ-H2AX (Fig. 2c, d). SMARCA5 recruitment was only visible in ~15% of the nuclei from either *BAZ1B*-KO cells or from *BAZ1A*-KO cells additionally depleted of BAZ1B by RNAi (BAZ1A KO, BAZ1B KD) (Fig. 2c, d; Supplementary Fig. 2c). Together, the data indicate that while BAZ1A and BAZ1B each promote the recruitment of SMARCA5 to chromatin lesions, BAZ1B plays a more critical role in this process.

**The bromodomain of BAZ1A bears a non-canonical gate-keeper**. To investigate whether BAZ1A may directly bind chromatin to localize SMARCA5 to DNA lesions, we examined the biochemical properties of putative chromatin-interacting domains within this ISWI regulatory subunit. The C-terminal regions of BAZ1A and BAZ1B each contain one BD of unknown function (Fig. 3a). Both BAZ1A-BD and BAZ1B-BD bear a conserved asparagine "anchor" residue that, in functional BDs, hydrogen bonds with the carbonyl group of a bound acyl-lysine

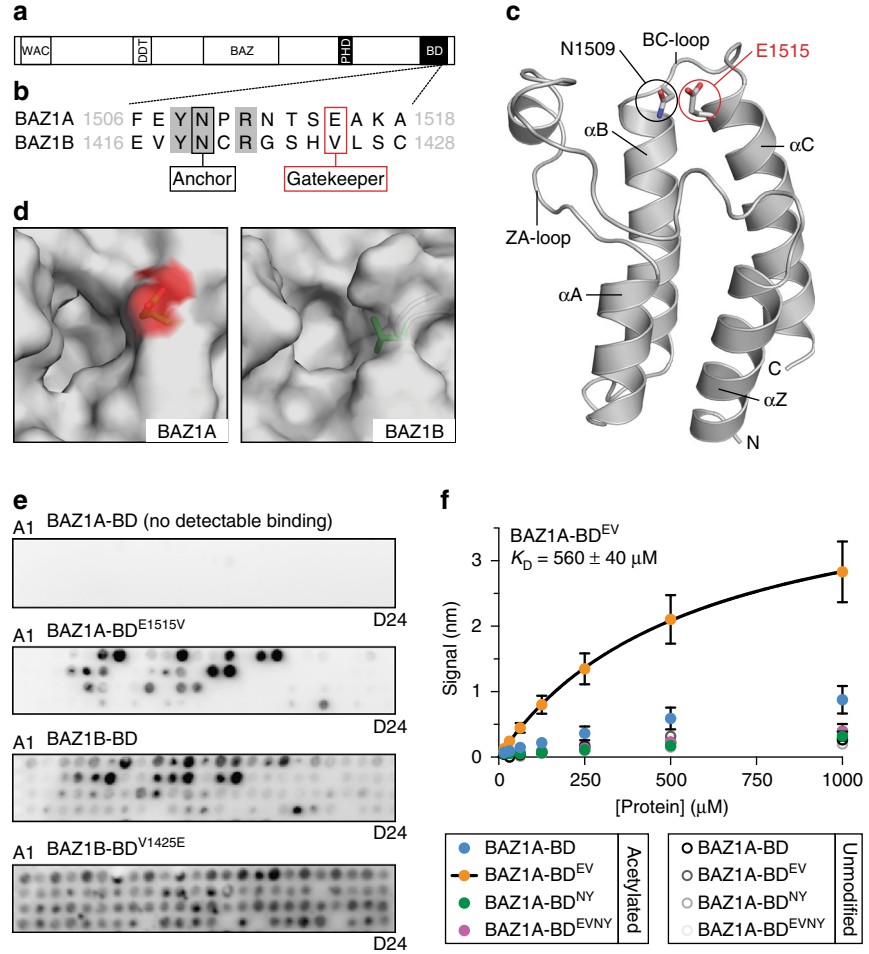

**Fig. 3** A non-canonical glutamic acid "gatekeeper" residue reduces the affinity of BAZ1A-BD for acetylated histone peptides. **a** Schematic representation of the BAZ1A and BAZ1B proteins. **b** Sequence of BAZ1A-BD and BAZ1B-BD surrounding the "anchor" and the "gatekeeper" residues; amino-acid boundaries are indicated. **c** Cartoon representation of BAZ1A-BD from the 1.7 Å resolution crystal structure (Table 1). **d** *Top* view of the binding pockets of BAZ1A-BD and BAZ1B-BD (model) in surface representation. The side chains of the gatekeeper residues are shown in *stick* representation with carbon atoms colored *green* and the oxygen atoms of the negatively charged BAZ1A-BD E1515 shown in *red*. **e** Representative examples of histone peptide array binding images for BAZ1A-BD, BAZ1A-BD$^{E1515V}$, BAZ1B-BD, and BAZ1B-BD$^{V1425E}$ protein modules. Refer to Supplementary Fig. 3c, d for peptide array information. **f** Biolayer interferometry binding study of wild-type BAZ1A-BD, BAZ1A-BD$^{E1515V}$ (BD$^{EV}$), BAZ1A-BD$^{N1509Y}$ (BD$^{NY}$), or the double mutant BAZ1A-BD$^{E1515V/N1509Y}$ (BD$^{EVNY}$). Steady-state binding was determined using biotinylated histone H4 1–19, either with or without acetylation of the four lysines present (see "Methods"). Each data point is the mean value ± s.e.m from four independent experiments. The fitted $K_D$ for BAZ1A-BD$^{E1515V}$ was determined from the averaged data and reported ± standard error

ligand and is required for this interaction (Fig. 3b)[28–30]. This suggests that BAZ1A-BD and BAZ1B-BD have the potential to bind an acetyl-lysine ligand. Additionally, BAZ1B-BD has a valine "gatekeeper" residue that is present in the binding pockets of many acetyl-lysine-binding BDs[29, 30]. However, the BAZ1A-BD gatekeeper is a non-canonical glutamic acid (Fig. 3b). We confirmed that this non-canonical feature is compatible with the BD fold by solving the crystal structure of BAZ1A-BD to 1.7 Å resolution (Fig. 3c; Table 1; Supplementary Fig. 3a). As expected, the glutamic acid gatekeeper introduces a negative charge on the side of the putative binding pocket of BAZ1A-BD (Fig. 3c, d; Supplementary Fig. 3b).

We used a peptide array to qualitatively probe the binding of recombinant BAZ1A-BD and BAZ1B-BD to a variety of acetylated histone peptides (Supplementary Fig. 3c, d)[30]. We could not detect BAZ1A-BD binding to any of the peptides on the array. In contrast, BAZ1B-BD readily bound to multiple acetylated peptides but not to unmodified peptides (Fig. 3e). To test whether the glutamic acid gatekeeper inhibits the binding of

BAZ1A-BD to acetylated peptides, we replaced it with a valine like in BAZ1B-BD. Unlike the wild-type domain, BAZ1A-BD$^{E1515V}$ showed substantial binding to the array that was dependent on peptide acetylation (Fig. 3e). Conversely, we found that replacing the canonical valine gatekeeper in BAZ1B-BD with a negatively charged glutamic acid appreciably reduced the binding of BAZ1B-BD$^{V1425E}$ to acetylated peptides and was accompanied by an increase in non-specific binding (Fig. 3e).

To better quantify peptide binding to BAZ1A-BD, we used biolayer interferometry to measure its interaction with a tetraacetylated H4 1–19 peptide. BAZ1A-BD binding was too weak to yield a meaningful $K_D$ determination; however, the binding signal was significantly higher than that seen with unmodified control peptide (Fig. 3f). In agreement with the array data, the E1515V substitution increased the binding of BAZ1A-BD$^{EV}$ to the acetylated peptide ($K_D = 560 \pm 40 \, \mu M$, $n = 4$; mean ± standard error) without increasing binding to the unmodified peptide (Fig. 3f). To test whether the weak binding of wild-type BAZ1A-BD to acetylated peptides was mediated by the canonical

**Table 1 X-ray data collection and refinement statistics**

| | BAZ1A BD |
|---|---|
| *Data collection* | |
| Space group | P1 |
| Cell dimensions | |
| *a, b, c* (Å) | 41.43, 44.92, 72.08 |
| *α, β, γ* (°) | 105.11, 89.95, 102.92 |
| Resolution (Å) | 1.70 (1.76–1.70)[a] |
| $R_{sym}$ or $R_{merge}$ (%) | 6.3 (54.0)[a] |
| $<I>/<\sigma I>$ | 19.6 (2.3)[a] |
| Completeness (%) | 95.7 (86.8)[a] |
| Redundancy | 1.9 (1.8)[a] |
| *Refinement* | |
| Resolution (Å) | 1.70 |
| No. of reflections | 52361 |
| $R_{work}/R_{free}$ (%) | 19.6, 23.5 |
| *No. of atoms* | |
| Protein | 3557 |
| Ligand/ion | 30 |
| Water | 240 |
| *Avg. B-factors* | |
| Protein | 25.4 |
| Ligand/ion | 21.4 |
| Water | 28.4 |
| *R.m.s. deviations* | |
| Bond lengths (Å) | 0.006 |
| Bond angles (°) | 0.967 |

All data for the structure were obtained from one crystal
[a]Values in parentheses are for highest resolution shell

BD pocket, we replaced the asparagine anchor with a tyrosine residue (N1509Y; BD[NY]) to completely abrogate acetyl-lysine recognition. The binding signal for the BAZ1A-BD[NY] mutant was indistinguishable from background and that observed for wild-type and unmodified peptides (Fig. 3f). Similarly, binding of BAZ1A-BD[EV] required the canonical asparagine anchor as the BAZ1A-BD[EVNY] double mutant failed to show increased affinity for acetylated vs. unmodified peptides (Fig. 3f). Poor solubility of BAZ1B-BD prevented the use of this protein module in biolayer interferometry or other biophysical assays. We conclude that, while the BDs of BAZ1A and BAZ1B are both compatible with canonical BD function, the presence of a glutamic acid at the gatekeeper position substantially lowers the binding affinity of BAZ1A-BD to acetylated histone peptides.

**The BDs of BAZ1A and BAZ1B aid DNA damage recovery**. Given that the BDs of BAZ1A and BAZ1B can bind acetyl-lysine, we asked whether this interaction was required for recovery from DNA damage. We used lentiviral transduction to stably introduce function-altering BD mutant versions of full-length BAZ1A and BAZ1B to *BAZ1A*-KO and *BAZ1B*-KO cells, respectively (Figs. 4a, 5a; Supplementary Figs. 4a, 5a). We re-expressed a version of BAZ1A that lacks the BD (1–1424; BAZ1A[ΔBD]), a mutant that promotes acetyl-lysine binding (E1515V; BAZ1A[EV]), and a substitution that abrogates acetyl-lysine recognition (N1509Y; BAZ1A[NY]). We attempted to re-express the double mutant (E1515V/N1509Y; BAZ1A[EVNY]), but low expression levels prevented us from evaluating this construct. Next, we monitored growth rate before and after a pulse of UV-induced DNA damage as described above (Fig. 1). We found that the complete BD deletion caused an increase in DNA damage sensitivity (Fig. 4b), suggesting an important role for the BD in damage recovery. In addition, defects were observed both for the gatekeeper substitution that enhances acetyl-lysine binding

(BAZ1A[EV]; Fig. 4c and Supplementary Fig. 4b, c) and, to a greater extent, for the inactivating mutation BAZ1A[NY] (Fig. 4d).

Having established that BAZ1A contributes to efficient recruitment of SMARCA5 to DNA lesions (Fig. 2), we monitored the effect of BAZ1A-BD mutations in this process. The slight DNA damage hypersensitivity of re-expressing BAZ1A[EV] cells was associated with a ~30% reduction in the recruitment of SMARCA5 to sites of DNA damage (Fig. 4e). In contrast, the removal (BAZ1A[ΔBD]) or complete inactivation (BAZ1A[NY]) of the BD of BAZ1A had no significant effects on SMARCA5 recruitment in *BAZ1A*-KO cells (Fig. 4e). The data indicate that acetyl-lysine recognition by BAZ1A-BD is not required for SMARCA5 accumulation at sites of damage and, additionally, that abnormally- high acetyl-lysine affinity (BAZ1A[EV]) may instead interfere with normal SMARCA5 localization.

BAZ1B-BD binds more readily to acetylated histone peptides compared to BAZ1A-BD (Fig. 3). We therefore asked whether acetyl-lysine recognition by BAZ1B-BD was required for survival or SMARCA5 localization to DNA lesions. We disrupted normal BAZ1B-BD function by substituting the canonical valine gate-keeper with a glutamic acid (V1425E; BAZ1B[VE]) or by removing the conserved asparagine anchor (N1419Y; BAZ1B[NY]) and monitored growth rate before and after a pulse of UVC light. We found that each mutation decreased cell proliferation after DNA damage (Fig. 5a–c and Supplementary Fig. 5a, b). However, like mutations that eliminate the binding of BAZ1A-BD to acetyl-lysine ligands, the BAZ1B[VE], and BAZ1B[NY] had no effect on SMARCA5 recruitment to DNA lesions (Fig. 5d). Together, the data indicate that binding of BAZ1A-BD and BAZ1B-BD to acetylated ligands promotes DNA damage recovery by an unidentified mechanism that is independent from SMARCA5 recruitment.

**The PHD module of BAZ1A binds to DNA**. BAZ1A contains a PHD of unknown function N-terminal to its BD (Fig. 3a). Many PHD modules recognize the N-terminal histone H3 tail, particularly when H3 is methylated[31, 32, 48]. However, BAZ1A-PHD lacks the "aromatic cage" and other key residues important for recognition of trimethylated lysine[49–51]. Moreover, BAZ1A-PHD lacks acidic residues that specifically recognize unmodified H3K4 peptides[52] (Supplementary Fig. 6a). Consistent with these observations, we detected no binding of recombinant BAZ1A-PHD to a commercial array encompassing a wide range of his-tone peptides (Supplementary Fig. 6b). Therefore, we investigated other biochemical properties that might allow BAZ1A-PHD to interact with chromatin.

BAZ1B-PHD has been structurally characterized[53]. BAZ1A-PHD is 54% identical and 73% similar to BAZ1B-PHD, and preserves the dual zinc-binding motif characteristic of the PHD fold (Fig. 6a). Accordingly, we used the structure of BAZ1B-PHD (PDB: 1F62) as template to model the three-dimensional structure of BAZ1A-PHD (Fig. 6b). By examining predicted surface electrostatic properties of the model, we noted a positively charged feature present in BAZ1A-PHD but missing in BAZ1B-PHD (Fig. 6b). This feature is generated by closely spaced lysines (K1181 and K1183; Fig. 6a) that could be involved in recognition of a negatively charged binding partner: one such possibility is DNA. We therefore assessed the binding of BAZ1A-PHD and BAZ1B-PHD to duplex DNA by biolayer interferometry. Steady-state measurements indicated that BAZ1A-PHD bound to DNA ($K_D = 51 \pm 9$ μM, $n = 10$), while BAZ1B-PHD did not bind (Fig. 6c). To test whether K1181 and K1183 are important for DNA binding, we mutated one or both to the corresponding residues in BAZ1B-PHD (A1227 and Y1229) (Fig. 6a). The single K1181A mutation severely reduced the signal of BAZ1A-PHD

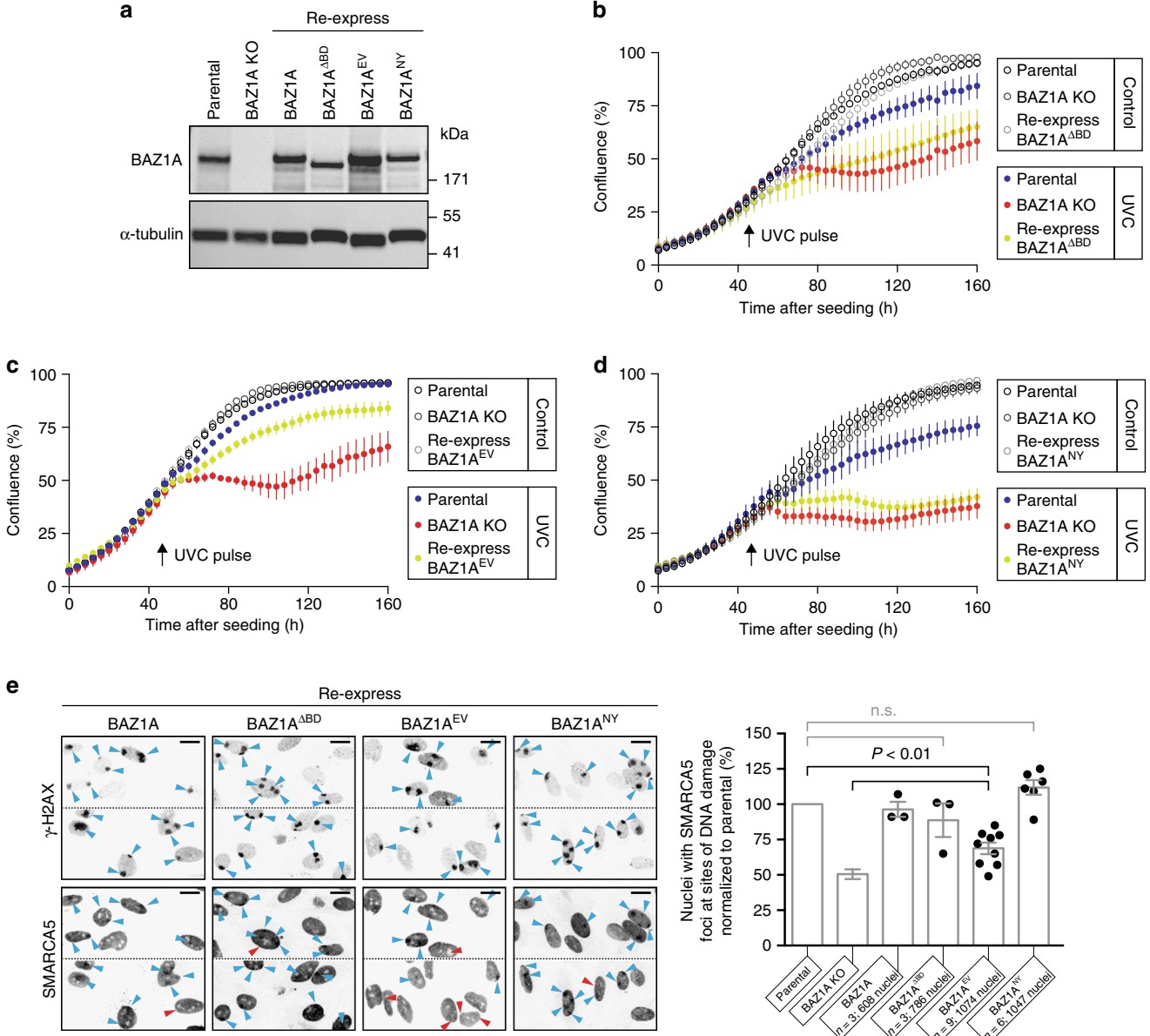

**Fig. 4** The BD of BAZ1A promotes DNA damage recovery but is not required for SMARCA5 recruitment. **a** Western blot analysis of BAZ1A expression in parental, *BAZ1A*-KO, and *BAZ1A*-KO cells engineered to re-express wild-type BAZ1A, or the mutants BAZ1A$^{\Delta BD}$ (1–1424), BAZ1A$^{EV}$ (E1515V), and BAZ1A$^{NY}$ (N1509Y). **b–d** Confluence of the indicated cell line measured over time, before, and after a pulse of UVC light. **e** Accumulation of endogenous SMARCA5 at sites of DNA damage as in Fig. 2c, d. *Scale bar*, 20 μm. Histograms represent the mean value ± s.e.m normalized to recruitment in parental cells. *P*-values ($P < 0.01$) were calculated using a two-tailed Mann–Whitney test; *n.s.* not significant ($P > 0.05$)

binding to DNA, and the BAZ1A-PHD K1181A/K1183Y double mutant no longer bound DNA (Fig. 6c). Similar results were obtained with reconstituted mononucleosomes, indicating that the histone octamer does not significantly inhibit the binding of BAZ1A-PHD to DNA (Fig. 6f and Supplementary Fig. 6h). To strengthen our observations, we measured the binding of BAZ1A-PHD to DNA by fluorescence polarization (Supplementary Fig. 6c, d). We used both direct and competition binding formats, and found that in solution BAZ1A-PHD readily binds DNA ($K_D = 5 \pm 1$ μM, $n = 3$), in a fashion dependent on the K1181 and K1183 residues. However, this pair of basic residues is not sufficient to confer DNA binding: neither BAZ1B-PHD A1227K/Y1229K nor the first PHD of *Drosophila* dAcf1 (dAcf1-PHD(1)), which has an equivalent lysine-arginine pair, is able to bind DNA (Supplementary Fig. 6e–g). These data are consistent with the

finding that the PHD modules of *Drosophila* dAcf1 instead bind to histone proteins[54] and suggest that additional residues of BAZ1A-PHD are involved in contacting DNA. Therefore, we used hydrogen-deuterium exchange mass spectrometry (HDX-MS) to map the DNA binding surface of BAZ1A-PHD. The relatively high deuterium ($^2$H) uptake for the N-terminal and C-terminal regions of uncomplexed BAZ1A-PHD is consistent with a globular core surrounded by more dynamic and solvent exposed N-termini and C-termini (Supplementary Fig. 7a, b). DNA binding considerably decreased $^2$H uptake toward the C-terminus of BAZ1A-PHD, encompassing K1181 and K1183, while the N-terminal region was less affected (Fig. 6d, e and Supplementary Fig. 7a, c). Thus, the HDX-MS data suggest that the DNA binding surface of BAZ1A-PHD is most likely contained within its C-terminal region.

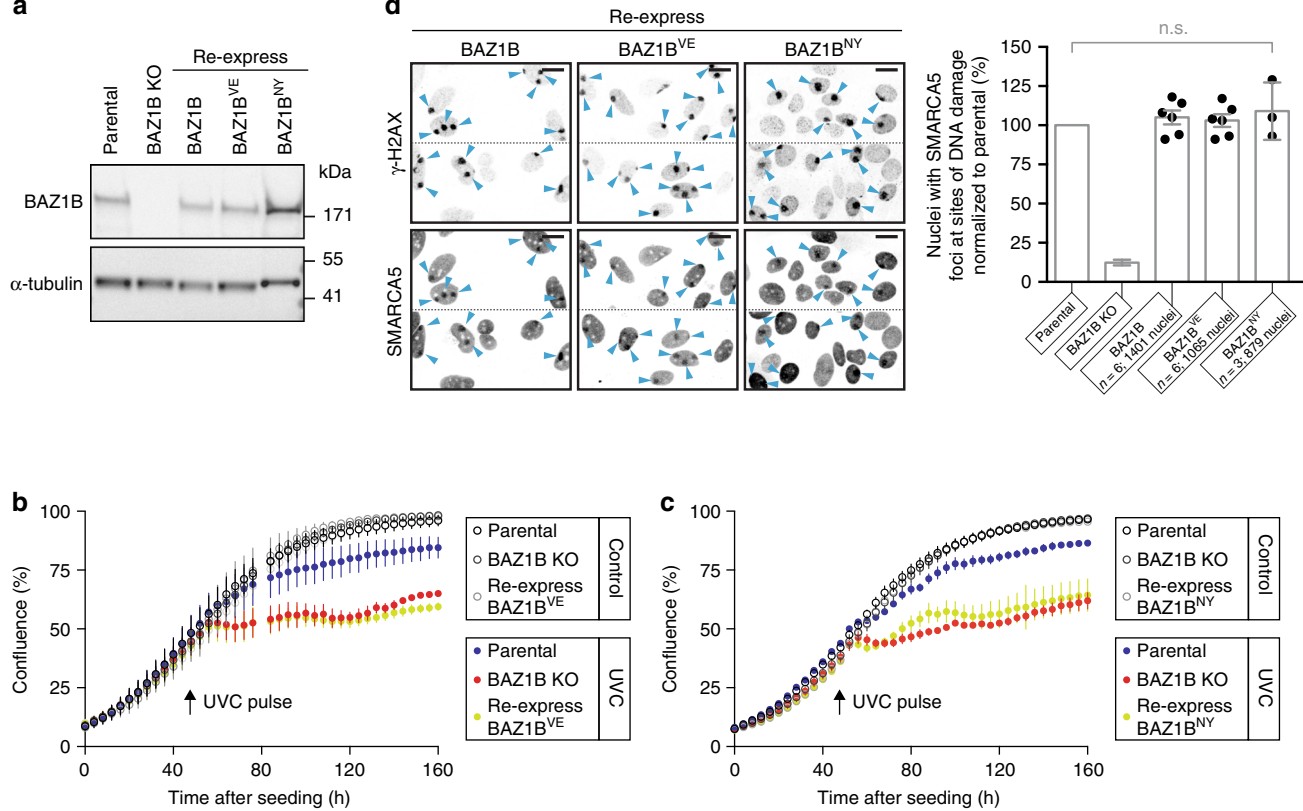

**Fig. 5** The BD of BAZ1B is important for DNA damage recovery but dispensable for SMARCA5 recruitment. **a** Western blot analysis of BAZ1B expression in parental, *BAZ1B*-KO and *BAZ1B*-KO cells engineered to re-express wild-type BAZ1B, or the mutants BAZ1B[VE] (V1425E) and BAZ1B[NY] (N1419Y). **b**, **c** Confluence of the indicated cell line measured over time, before, and after a pulse of UVC light. **d** Accumulation of endogenous SMARCA5 at sites of DNA damage as in Fig. 2c, d. *Scale bar*, 20 μm. Histograms represent the mean value ± s.e.m normalized to recruitment in parental cells. *P*-values were calculated using a two-tailed Mann–Whitney test; *n.s.* not significant ($P > 0.05$)

The N-terminal WAC motif of BAZ1A was shown to bind nucleosomal linker DNA to regulate the activity of the BAZ1A-SMARCA5 (ACF) complex[55]. To test whether the PHD–DNA interaction reported here might play a similar role, we co-expressed full-length human BAZ1A, or the BAZ1A[KAKY] mutant (K1181A/K1183Y) with SMARCA5 in insect cells and isolated highly pure ACF and ACF[KAKY] complexes (Supplementary Fig. 7d). This shows that the BAZ1A[KAKY] mutant is stably incorporated into a complex with SMARCA5. Next, we used mononucleosomes as remodeling substrate and monitored the sliding of the octamer from the edge toward more central positions through a change in electrophoretic mobilities[56]. Consistent with previous data, SMARCA5 remodeled homogenously edge-positioned nucleosomes toward a mixed population of edge and central configurations, while the ACF complex preferentially remodeled mononucleosomes to the center[56] (Supplementary Fig. 7e). Similarly, ACF[KAKY] readily centered mononucleosomes in an ATP-dependent fashion, indicating that, unlike the WAC–DNA interaction, the PHD–DNA interaction discovered here is not required for this process.

**PHD mutations alter BAZ1A recruitment and gene expression.** Having established that BAZ1A contributes to the DNA damage response, we sought to determine whether the unusual DNA-binding property of its PHD module might contribute to the cellular function of BAZ1A. Therefore, we engineered into full-length BAZ1A PHD point mutations that disrupt the PHD–DNA interaction (BAZ1A[KAKY]; Fig. 6). Protein levels of BAZ1A[KAKY] stably re-expressed in *BAZ1A*-KO cells were comparable to BAZ1A in parental cells (Fig. 7a). Next, we monitored growth

rate before and after UV-induced DNA damage as described above (Fig. 1). To strengthen our observations, we cloned and evaluated two individual BAZ1A[KAKY] mutant cell lines (Fig. 7b and Supplementary Fig. 8a, b). BAZ1A with a PHD incapable of binding DNA failed to rescue the DNA damage hypersensitivity of *BAZ1A*-KO cells. The defect of the BAZ1A[KAKY] mutant in this assay resembled that of the complete KO (Fig. 7b; Supplementary Fig. 8b), demonstrating that the function of BAZ1A-PHD is an important aspect of the function of full-length BAZ1A in responding to DNA damage.

The role of BAZ1A-PHD might be to recruit SMARCA5 to chromatin lesions through direct binding to DNA. However, like the inactivation of BAZ1A-BD (Fig. 4, BAZ1A[NY]), the BAZ1A[KAKY] mutant did not appreciably affect the recruitment of SMARCA5 to DNA damage (Fig. 7c). Next, we investigated whether the PHD module might more subtly regulate the interaction of BAZ1A with damaged chromatin by comparing the recruitment of GFP-BAZ1A and GFP-BAZ1A[KAKY] in live cells. Fluorescence intensity was measured after laser-induced damage (and concomitant GFP photobleaching) and compared to a non-irradiated site. Bleaching of the GFP signal causes the initial irradiated-over-control ratio to be <1, whereas signal >1 indicates GFP-BAZ1A enrichment in response to damage (Fig. 7d). We found that, compared to wild-type BAZ1A, the signal for the BAZ1A[KAKY] mutant recovered ~40% faster after photobleaching, and BAZ1A[KAKY] accumulation at sites of damage was ~20% increased (Fig. 7d). This effect was only seen for the PHD mutant: BD mutant BAZ1A[EV] closely resembled wild-type BAZ1A in this assay (Supplementary Fig. 8c).

In addition to physically remodeling chromatin at sites of DNA damage, BAZ1A (and the ACF complex) might promote recovery from DNA lesions indirectly by regulating gene expression[33, 57, 58]. Consistent with this notion, only a fraction of the BAZ1A protein present in the nucleus is recruited to damaged chromatin

(Fig. 2a; ref.[16]). Therefore, we transcriptionally profiled (using RNA sequencing) parental and *BAZ1A*-KO cells, as well as cells engineered to re-express BAZ1A wild-type and the BAZ1A[KAKY] PHD mutant, both before and after UV-induced DNA damage. Principal component (PC) analysis revealed that UVC irradiation

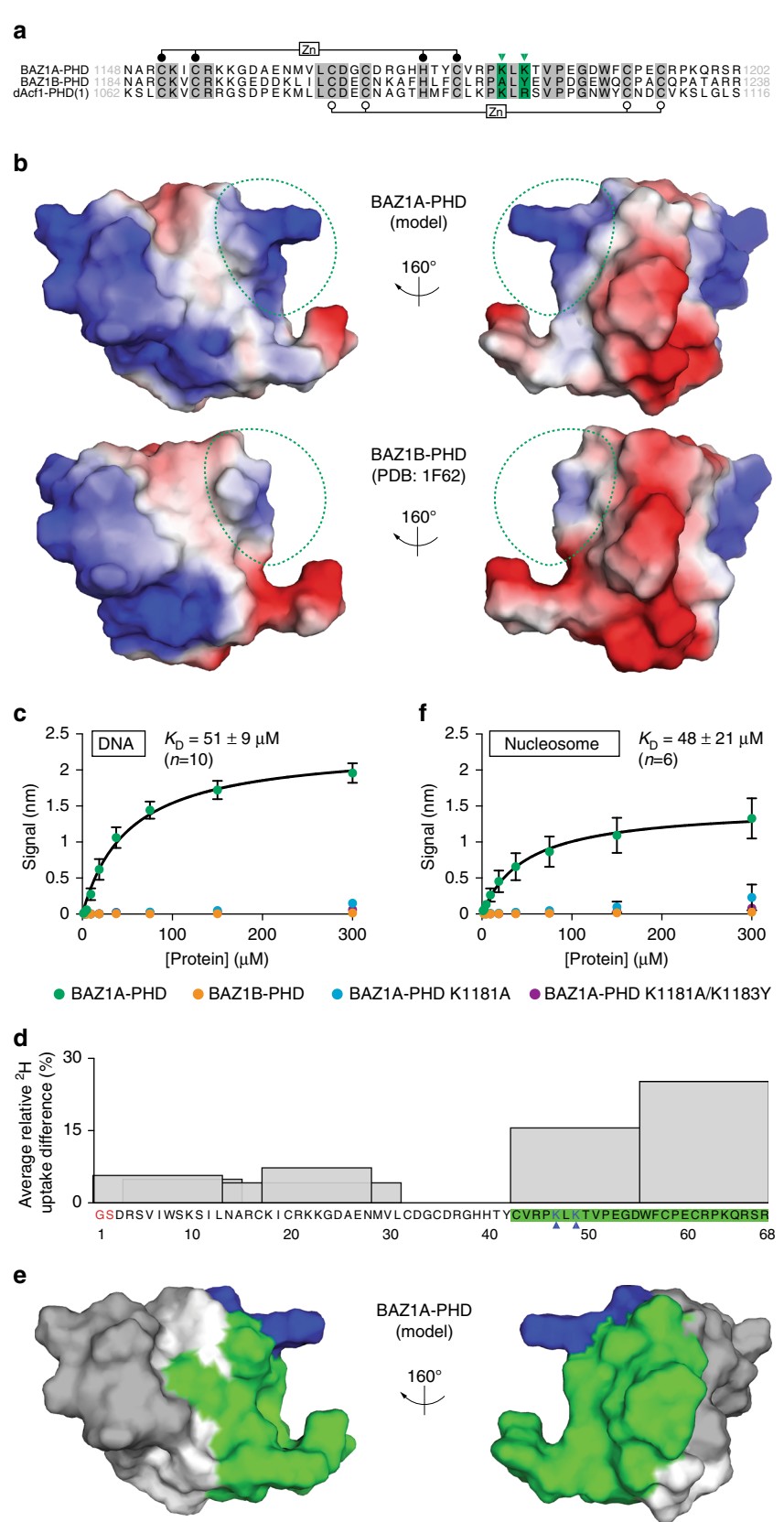

alters gene expression in all cell lines tested, indicating that BAZ1A is not generally required to elicit a transcriptional response to DNA damage (Fig. 7e). To better characterize the DNA damage hypersensitivity of the *BAZ1A*-KO and BAZ1A-KAKY cells, we compared the expression profiles of parental and re-expressing BAZ1A cells as well as *BAZ1A*-KO and BAZ1A-KAKY cells, before and after UV-induced DNA damage. We found a handful of genes that were specifically induced in parental and re-expressing BAZ1A cells but not in *BAZ1A*-KO or BAZ1A-KAKY cells (Fig. 7f). These included SPINK13, a potential serine protease inhibitor, and several poorly characterized transcripts (see "Discussion"). Moreover, several genes were more highly expressed in cells with wild-type BAZ1A compared to *BAZ1A*-KO and BAZ1A-KAKY cells, even before DNA damage induction (Fig. 7f). However, consistent with a role for BAZ1A in transcriptional repression[33, 57, 58], an even larger number of genes had lower expression in parental and re-expressing BAZ1A cells compared to BAZ1A mutant cells, both before and after irradiation (Supplementary Fig. 8d).

To investigate the transcriptional dysregulation caused by the BAZ1A-KAKY mutant in particular, we performed a 4-way comparison of re-expressing BAZ1A wild-type and BAZ1A-KAKY mutant cells in control and UVC-irradiated conditions (Fig. 7g; Supplementary Data 1). Re-expressing BAZ1A wild-type and BAZ1A-KAKY cells provide the best comparison given that both lines are clonal and were produced following exactly the same procedure. Consistent with the significant derepression caused by genetic removal of BAZ1A (Supplementary Fig. 8d; ref.[33]), a total of 1141 genes were significantly upregulated in BAZ1A-KAKY vs. wild-type cells either before or after DNA damage, and only 421 were downregulated. Among the upregulated set, 306 genes were specifically derepressed after DNA damage (Fig. 7g). Enrichment for pathways derived from the Reactome database showed that genes abnormally induced in BAZ1A-KAKY cells in response to DNA damage belong to signaling pathways that regulate cell differentiation, proliferation, and apoptosis (see Supplementary Data 1). Together, this indicates that a version of BAZ1A that is unable to bind DNA via its PHD module is abnormally recruited to sites of DNA damage and causes transcriptional dysregulation.

## Discussion

Chromatin remodelers of the ISWI family have emerged as regulators of the DNA damage response[11]. Here, we use CRISPR-Cas9-mediated and lentiviral-mediated genome editing to reveal that the non-catalytic subunits BAZ1A and BAZ1B are key for recovery after DNA damage in human cells, in part by recruiting the ATPase SMARCA5 to damaged chromatin. This finding is in disagreement with mouse *Baz1a* KO studies that do not support a role for BAZ1A in DSB repair[33]. It is unlikely that the effects seen here are caused by a specific requirement for BAZ1A in tumor-derived cells, given that earlier studies implicated BAZ1A in responding to DNA damage in primary human cells[17, 22].

One possibility is that the role of BAZ1A in responding to DNA damage is more critical in humans than it is in mice. Mendelian ratios of *Baz1a*⁻/⁻ offspring were recovered from heterozygous crossings in mice, suggesting that the rise of compensatory mutations might not be required for viability[33]. However, an alternative possibility is that non-mutational developmental compensation might occur in the KO mice. While BAZ1B expression was no higher in ear fibroblasts derived from *Baz1a*⁻/⁻ mice[33], it remains possible that BAZ1B, or other ISWI regulatory subunits, may functionally compensate for the absence of BAZ1A in mice.

Our study reveals that the PHD and BD domains of BAZ1A each support BAZ1A cellular function through the non-canonical biochemical features discovered here. BAZ1A-BD bears a glutamic acid gatekeeper that reduces its binding affinity to acetylated histone peptides. We find that substitution of this evolutionarily conserved glutamic acid with a more canonical valine increases binding to acetylated histone peptides and causes modest DNA damage hypersensitivity. Moreover, the complete elimination of acetyl-lysine recognition by removal of the asparagine anchor is highly detrimental to the function of BAZ1A in cells. Together, the data suggest that BAZ1A-BD has evolved to bind acetylated histones with a specific affinity, and either increasing or decreasing the affinity of this interaction prevents BAZ1A from functioning in DNA damage recovery. However, we cannot exclude that BAZ1A-BD has a distinct, non-canonical function that is impaired by the removal of the glutamic acid gatekeeper.

A large number of studies have shown that PHD modules function by binding to unmodified or post-translationally modified histones and non-histone proteins[31, 32]. In contrast, the second PHD of human BRD1 binds DNA in a non-specific manner[59]. Here, we show that the BAZ1A-PHD is a second example of a PHD module that can bind DNA, suggesting that DNA binding might be functionally relevant more broadly for PHD modules. It is remarkable that two point mutations in BAZ1A-PHD (BAZ1A-KAKY) that abrogate DNA binding cause DNA damage hypersensitivity in growth assays that is very similar to that seen for *BAZ1A*-KO cells, yet SMARCA5 recruitment is unaffected. Using live-cell imaging we show that the BAZ1A-KAKY hyper-accumulates at sites of damage. These data are consistent with a previous study showing that the PHD

---

**Fig. 6** The PHD module of BAZ1A binds free and nucleosomal DNA. **a** Sequence alignment of the PHD modules of BAZ1A, BAZ1B, and the most N-terminal PHD module of dAcf1 amino acid boundaries are indicated, and identical residues are highlighted in *gray*. *Filled* and *open circles* indicate the Zn-binding Cys$_4$HisCys$_3$ motif characteristic of the PHD fold. *Green arrowheads* indicate K1181 and K1183 that are required for BAZ1A-PHD to bind DNA. **b** Electrostatic surface representation of BAZ1A-PHD (model) and BAZ1B-PHD (PDB: 1F62). *Red* and *blue* indicate negatively and positively charged areas, respectively. The positively charged surface formed by K1181 and K1183 in the BAZ1A-PHD model and the corresponding region in BAZ1B-PHD are outlined in *dotted green*. **c** Steady-state biolayer interferometry measurement of DNA binding to wild-type BAZ1A-PHD, BAZ1B-PHD, or mutant versions of BAZ1A-PHD bearing the single K1181A or the double K1181A/K1183Y substitutions. **d** Average relative deuterium uptake difference (ARDD) for BAZ1A-PHD bound to DNA compared to uncomplexed BAZ1A-PHD, measured by HDX-MS. The values for six matching peptides (recovered from both DNA-bound and uncomplexed samples) are represented by histograms spanning the corresponding amino acid sequence of the BAZ1A-PHD construct. No matching peptides were obtained for residues 32–42 and histograms of overlapping peptides are shown in transparency. Residue numbering refers to the tag-free expression construct used for this experiment; residues colored in *red* do not belong to the endogenous BAZ1A sequence. The two lysine residues colored in *blue* and highlighted with *blue arrowheads* correspond to K1181 and K1183. Peptide sequences with the highest ARDD values (>10%) are highlighted in *green*; see Supplementary Fig. 7a, and the "Methods" section for more details. **e** Peptides with the highest ARDD values (>10%) were mapped on the BAZ1A-PHD model surface and shown in *green*. K1181 and K1183 (residues 47 and 49 in **d**) are colored in *blue*, and *white* areas correspond to residues 32–42, for which no ARDD data was obtained. **f** Same as **c** but using mononucleosomes. Each data point is the mean value ± s.e.m from four independent experiments. Fitted $K_D$s were determined from the averaged data and reported ± standard error

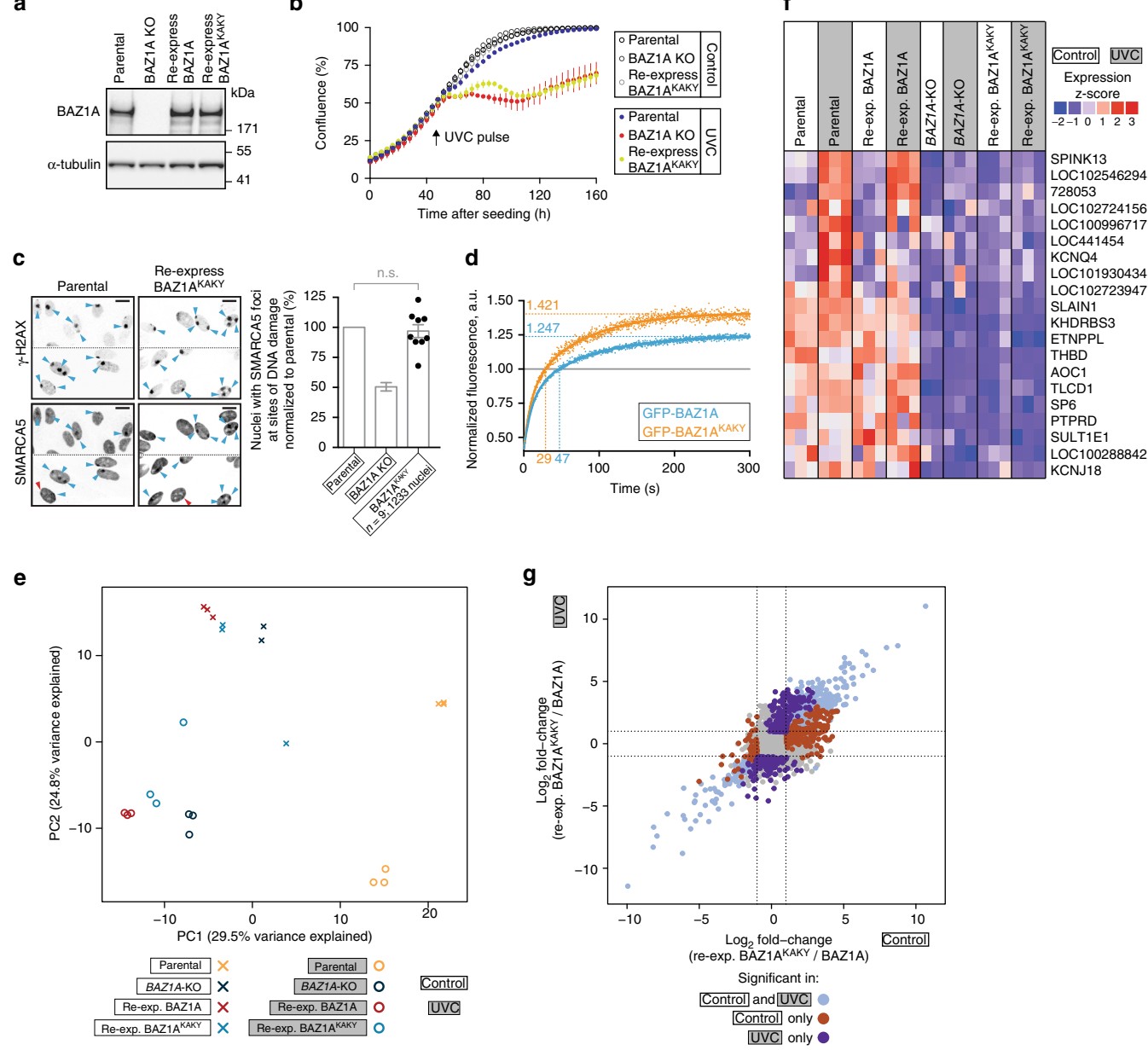

**Fig. 7** Function-altering PHD substitutions are detrimental to the cellular functions of BAZ1A. **a** Western blot analysis of BAZ1A expression in parental, *BAZ1A*-KO, and *BAZ1A*-KO cells engineered to re-express BAZ1A$^{KAKY}$. **b–d** Confluence of the indicated cell line measured over time, before, and after a pulse of UVC light. **c** Accumulation of endogenous SMARCA5 at sites of DNA damage as in Fig. 2c, d. *Scale bar*, 20 μm. Histograms represent the mean value ± s.e.m. normalized to recruitment in parental cells. *P*-values were calculated using a two-tailed Mann–Whitney test; *n.s.* not significant (*P* > 0.05). **d** Accumulation of GFP-fusion BAZ1A proteins to sites of laser irradiation in live cells. The fluorescence signal at site of irradiation is normalized to the signal of a non-irradiated area. Signal above 1 indicates accumulation at site of damage. Numbers on the *x*-axis and *y*-axis represent the time at which the fluorescence value equals 1 and the maximum value (plateau determined by fitting), respectively. **e** PC analysis of the transcriptional profiles of parental, *BAZ1A*-KO, re-expressing BAZ1A wild-type and re-expressing BAZ1A$^{KAKY}$ cells, before and after DNA damage. For each cell line, the first and second PCs (explaining the highest percent of variance) are reported for 2–3 independent RNA extractions. **f** Heat map showing the expression profiles of genes that are significantly upregulated in parental and re-expressing BAZ1A cells after UVC treatment compared to BAZ1A$^{KAKY}$ and *BAZ1A*-KO cells. The genes that are significantly downregulated for the same comparison are shown in Supplementary Fig. 8d. In both cases, only genes with the same expression profiles in parental and re-expressing BAZ1A cells, as well as *BAZ1A*-KO and BAZ1A$^{KAKY}$ cells are reported. Individual columns correspond to independent RNA extractions. **g** 4-way comparison showing the log$_2$ fold-change between *BAZ1A*-KO cells engineered to re-express BAZ1A wild-type or the BAZ1A$^{KAKY}$ mutant under control conditions on the *x*-axis compared to UVC conditions on the *y*-axis. Points are colored according to differential expression significance, points in *gray* do not satisfy significance cut-off (*P* < 0.01, absolute log$_2$ fold-change >1)

module is not required for BAZ1A recruitment at DNA lesions[15]. Moreover, it is unlikely that the general DNA binding activity uncovered here would be required for specific targeting of BAZ1A to sites of DNA damage. Instead, the DNA interaction surface of

BAZ1A-PHD might more generally regulate BAZ1A distribution in the nucleus and could be required for disengaging BAZ1A from chromatin after DNA repair has been completed; this hypothesis remains to be tested.

PHD mutations that disrupt DNA binding and the genetic removal of BAZ1A each cause transcriptional dysregulation in human cells both before and after DNA damage. Among a handful of poorly characterized genes, SPINK13 is specifically upregulated in response to DNA damage in cells bearing wild-type BAZ1A but is not expressed in BAZ1A mutant cells. Given that SPINK13 has been implicated in sperm maturation[60], this observation provides a possible link between the DNA damage hypersensitivity reported here for BAZ1A-KO cells and the striking defects in sperm development observed in Baz1a−/− mice[33].

While genetic removal of BAZ1A and BAZ1B hinders the recruitment of SMARCA5 to sites of DNA damage, multiple point mutations within these ISWI regulators cause DNA damage hypersensitivity without affecting SMARCA5 localization. These data strongly suggest that SMARCA5 recruitment is not sufficient to promote recovery from DNA damage. BAZ1A, BAZ1B, and SMARCA5 have been implicated in multiple distinct DNA repair processes[11]. Thus, it is likely that fully functional regulatory subunits are required in complex with SMARCA5 to play a fundamental role in overcoming DNA damaging events. One possibility is that these ISWI factors remodel chromatin around damaged DNA to increase local DNA accessibility and promote repair. Alternatively, they may function to reassemble histone-DNA contacts after the repair as been completed. The molecular interactions uncovered here suggest some ways forward to define the exact role of ISWI remodelers in responding to DNA damage.

## Methods

**Cell line generation**. For CRISPR-Cas9 genome editing a human codon-optimized cDNA encoding *S. pyogenes* Cas9 was cloned downstream of the human cytomegalovirus immediate-early promoter in a pRK5 vector backbone. Individual small guide RNAs (sgRNAs) for human BAZ1A and BAZ1B were cloned downstream of the human U6 promoter of the pLKO.5 vector (Sigma). See references in the text and Supplementary Fig. 1 for the sequences of each sgRNA. Cas9 and sgRNA constructs were co-transfected in in-house HeLa cells (originally obtained from ATCC) with the Lipofectamine 2000 reagent (Thermo Fisher Scientific). Three days post-transfection, cell were harvested, and plated in 96-well plates at a dilution of 0.5 cell per well. Clonal colonies were expanded and tested by Western blot with anti-BAZ1A (A301-318A, Bethyl Laboratories), anti-BAZ1B (NB600-279, Novus Biologicals), and anti-Tubulin (T9028, Sigma) antibodies. For concurrent depletion of BAZ1B in BAZ1A-KO cells, the DSIR[61] algorithm was used to design a BAZ1B-specific siRNA that was embedded into DNA oligonucleotides and used to generate a shRNA expression cassette in the pINDUCER10 backbone[62]. The targeting sequence of BAZ1B shRNA is TAG GTG CAA AGT TTG TCG AAA. To induce BAZ1B knockdown, cells were grown in the presence of 500 ng $\mu L^{-1}$ doxycycline. For re-expression of BAZ1A wild-type, as well as the BAZ1A$^{EV}$ (E1515V) and BAZ1A$^{KAKY}$ (K1181A/K1183Y) mutants in BAZ1A-KO cells, and BAZ1B wild-type and the BAZ1B$^{VE}$ (E1425V) mutant in BAZ1B-KO cells, the corresponding cDNAs were synthesized (GenScript) and cloned into the pLENTI6.3 backbone (Thermo Fisher Scientific). Lentiviruses were generated by co-transfecting $6 \times 10^6$ 293T cells with the packaging system composed of pCMV-VSVG and pCMV-dR8.9 plasmids and shRNA/cDNA-containing vectors (pINDUCER10 or pLENTI6.3). The supernatant was harvested after 3 days, and lentiviruses were concentrated 10-fold using Lenti-X Concentrator (Clontech). Concentrated lentiviruses were supplemented with 8 $\mu l$ $mL^{-1}$ polybrene (EMD Millipore), and $0.6 \times 10^6$ HeLa cells were spin-infected at room temperature for 45 min. Cells were grown for 3 days before being expanded and maintained in presence of 10 $\mu g$ $mL^{-1}$ blasticidin or 2 $\mu g$ $mL^{-1}$ puromycin. For re-expression of BAZ1A$^{\Delta BD}$ (1–1424), BAZ1A$^{NY}$ (N1509Y), and BAZ1A$^{EVNY}$ (E1515V/N1509Y) in BAZ1A-KO cells, and BAZ1B$^{NY}$ (N1419Y) in BAZ1B-KO cells, as well as expression of GFP-fusion constructs in BAZ1A-KO cells, a piggyBac transposon system was used. The piggyBac transposase was supplied via the pBO vector (Transposagen). The cDNAs for BAZ1A and BAZ1B constructs were generated by custom gene synthesis and standard restriction cloning using a pUC57 vector backbone (GenScript). All cell lines were mycoplasma-free and their identity was confirmed by single-nucleotide polymorphism profiling. See Supplementary Fig. 9 for source images from which Western blot figures were extracted.

**Cell culture conditions and treatments**. HeLa parental and genome-edited cell lines were maintained in Dulbecco's Modified Eagle Medium, supplemented with 10% fetal bovine serum and 2 mM L-glutamine at 37 °C and 5% $CO_2$. For UVC pulse treatment, cells were seeded in 96-well plates with 100 $\mu L$ medium, and, after attachment overnight, confluence was monitored at 4 h-intervals in an IncuCyte ZOOM instrument (Essen BioScience) maintained at 37 °C and 5% $CO_2$. 48 h after

seeding, cells were exposed to a UVC pulse ($\lambda = 254$ nm, 100 J $m^{-2}$) in a Hoefer UVC 500 Ultraviolet Crosslinker, and confluence was monitored for a total of 160 h. Time points were averaged from eight technical replicates. Control wells on the same plate were exposed to the UVC pulse without removing the plastic cover. For treatment with phleomycin D1, Zeocin (Thermo Fisher Scientific) was added at seeding to a final concentration of 20 $\mu M$. When monitoring cell death, SYTOX Green nucleic acid stain (Thermo Fisher Scientific) was added at seeding to a final concentration of 5 $\mu M$, and images were taken simultaneously in phase-contrast and green fluorescence modes on an IncuCyte FLR instrument. SYTOX Green nucleic acid stain only penetrates compromised membranes characteristic of dying cells. Green fluorescence was normalized to cell confluence, and the ratio of normalized fluorescence in the absence or in the presence of phleomycin D1 is reported.

**Laser micro-irradiation and microscopy**. For laser micro-irradiation and UVC irradiation through a porous membrane, HeLa cells were grown for 3 days in the presence of 10 $\mu g$ $mL^{-1}$ BrdU as a photosensitization agent. Laser micro-irradiation in living cells was induced using a Nikon A1R laser-scanning confocal microscope (Nikon Instruments) equipped with a Tokai Hit stage-top environmental incubator (Tokai Hit). Cells were grown on glass-bottom 24-well plates (E&K Scientific), and prior to irradiation the medium was removed and replaced with Hank's balanced salt solution (HBSS). The Elements software (Nikon Instruments) was used to draw a $11.6 \times 1.9$ $\mu m$ region of interest (ROI) stimulation area within the nucleus of two to three cells in each field of view. Typically three fields were selected in each well. DNA damage was induced in each ROI by irradiation using a 405 nm laser at 100% power, set to scan each ROI 1000 times at a scan speed of 1/8 fps. Within 2 h after irradiation, plates were removed from the microscope and processed for immunofluorescence. Cells were fixed with 4% paraformaldehyde in HBSS for 30 min at room temperature, permeabilized with 0.05% Triton X-100 in HBSS, and blocked in 5% bovine serum albumin, 0.5% Teleostean Gelatin, and 0.05% Tween-20 in HBSS. Cells were stained with mouse anti-γH2AX (05-636, Sigma) and Alexa Fluor 647 goat anti-mouse (A31626, Thermo Fisher Scientific), followed by rabbit anti-SMARCA5 (NB100-55310, Novus Biologicals), anti-BAZ1A (NB100-61041, Novus Biologicals), or anti-BAZ1B (NB600-279, Novus Biologicals) and Alexa Fluor 488 goat anti-rabbit (A31628, Thermo Fisher Scientific). Primary and secondary antibodies were used at 1/200 and 1/500 dilutions, respectively. After staining, confocal images were obtained by returning the plate to the Nikon A1R confocal microscope, collecting Z-stacks through each nucleus using a 20 × Plan Apo water-immersion objective, and processing images as maximum intensity projections (Nikon Elements). Differential interference-contrast images were collected before and after irradiation using the same objective. For quantitative analysis of SMARCA5 accumulation at sites of damage, the medium was removed, cells were briefly washed with HBSS, and immediately subjected to UVC irradiation ($\lambda = 254$ nm, 200 J $m^{-2}$) in a Hoefer UVC 500 Ultraviolet Crosslinker through a 5 $\mu m$ porous membrane (TMTP02500, Millipore) that was directly placed on top of the cell layer[47]. Membranes were removed by addition of fresh medium and cells were processed for imaging as above. In parental cells, SMARCA5 foci were typically found in ~70% of cells. Live-cell imaging and fluorescence recovery after photobleaching was carried out on a Nikon A1R confocal microscope with 40 × /1.15 NA Water Immersion objective using Nikon Elements software (Nikon Instruments, New York). The microscope was equipped with a stage-top environmental chamber set to 37 °C and humidified at 5% $CO_2$ (Tokai Hit). DNA damage and fluorophore bleaching were performed using the 405 nm diode laser at 100% power to scan a ROI ($19 \times 2$ $\mu m$) with 500 iterations at a scan speed of 8 fps. Cells were simultaneously imaged at 0.2 s intervals with the 488 nm laser during and after bleaching, for a total of 5–10 min. The normalized fluorescence reported in Fig. 7d corresponds to the ratio of GFP signal intensity at the ROI over a non-irradiated area of the same nucleus, immediately following laser irradiation. Each point shown is the mean value ± s.e.m. from at least nine independent experiments. The data were fitted to a two-phase exponential decay distribution.

**Recombinant protein production in *Escherichia coli***. Coding sequences for wild-type and mutant BD and PHD modules were synthesized (Integrated DNA Technologies) and cloned into one of two expression vectors: either a modified pRSF (Novagen) or pET52b (Novagen). Alternatively, mutant constructs were produced by site-directed mutagenesis (Agilent Genomics). Proteins were expressed as N-terminal His$_6$-FLAG-tagged fusions with TEV or thrombin protease sites for removal of tags. The domains spanned the following residues: BAZ1A-BD (S1425–T1538), BAZ1B-BD (S1337–V1450), BAZ1A-PHD (D1137–R1202), BAZ1B-PHD (D1172–R1237), and dAcf1-PHD(1) (H1051–L1115). Constructs were transformed into Rosetta2(DE3) cells (Novagen), cells were grown for 4 h with shaking at 37 °C, and proteins were expressed for 20 h at 18 °C by auto-induction[63]. After harvesting by centrifugation, cell pellets were resuspended in 50 mM Tris pH 7.5, 300 mM NaCl, 1 mM tris(2-carboxyethyl)phosphine (TCEP), 10% glycerol, 10 mM imidazole, and 0.2% Tween-20, in the presence of protease inhibitors (cOmplete, EDTA-free, Roche) and 5 U $mL^{-1}$ Benzonase Nuclease (Sigma). Cells were lysed by sonication or by passage through a M-110Y Microfluidizer (Microfluidics). The cleared lysates were applied onto Ni-NTA Superflow resin (Qiagen) in batch mode for 1 h at 4 °C. The resin was moved to a gravity column and washed extensively with 50 mM Tris pH 7.5, 300 mM NaCl, 1 mM

TCEP, 10% glycerol, and 10 mM imidazole. Proteins were then eluted with 50 mM Tris pH 7.5, 300 mM NaCl, 1 mM TCEP, 10% glycerol, and 300 mM imidazole. For peptide-array binding assays, Ni-NTA elution fractions were concentrated using Amicon Ultra centrifugal filter units with 3 kDa cut-off (Millipore) and subjected to size-exclusion chromatography (SEC) on a Superdex S75 preparative column (HiLoad 16/6, GE Healthcare), pre-equilibrated with 20 mM Tris pH 7.5, 200 mM NaCl, 10% glycerol, and 1 mM TCEP. For biolayer interferometry measurements, tags were removed with the appropriate protease, and the samples were applied again onto Ni-NTA Superflow resin (Qiagen). The solution containing the unretained, untagged proteins was concentrated and subjected to SEC as described above. All protein samples were pooled from a single SEC peak of the expected retention volume, and purity (>95%) was confirmed by SDS-PAGE. Proteins were concentrated as described above, and concentration was determined by absorbance at 280 nm with extinction coefficients calculated using ProtParam (ExPASy, http://web.expasy.org/protparam/). Aliquots were flash frozen in liquid nitrogen, and stored at −80 °C.

**Structure determination and homology modeling.** Purified, untagged BAZ1A-BD was exchanged into 20 mM HEPES pH 7.5, 150 mM NaCl, 1 mM TCEP using PD-10 desalting columns (GE Healthcare Life Sciences) and concentrated to 29.9 mg ml$^{-1}$. Crystals were obtained by mixing equal volumes of protein and well solutions, using sitting-drop vapor diffusion at 19 °C. The well solution contained 100 mM HEPES pH 7.2 and 25% PEG 3350. Plate-shaped crystals were flash frozen in well solution supplemented with 20% glycerol. Data were collected at ALS 502 and processed using HKL2000[64]. The beam used had a wavelength of 0.9795 Å and the temperature for data collection was 100 K. The structure of yeast GCN5 bromodomain (PDB: 1E6I) was used as a model for molecular replacement, and a solution was obtained using PHASER[65] in the CCP4 Program Suite. The asymmetric unit contained four molecules of BAZ1A-BD. The models were then adjusted using COOT[66] to F$_O$-F$_C$ and 2F$_O$-F$_C$ maps, and the structure was refined using PHENIX[67]. Addition of HEPES and water molecules amid cycles of building and refinement produced the final model (Table 1). The final model had 100% of the residues in most favored regions of the Ramachandran plot and only 0.5% rotamer outliers. Homology modeling was performed using the Homology Modeler application from the Molecular Operating Environment package (MOE 2012.10, Chemical Computing Group). The experimental crystal structure of BAZ1A-BD determined here was used as the template to model BAZ1B-BD. The experimental crystal structure of BAZ1B-PHD (PDB: 1F62; ref. [53]) was used as the template for BAZ1A-PHD and dAcf1-PHD(1) models. Qualitative surface electrostatic distribution was predicted using the Protein Contact Potential representation in PyMOL (PyMOL 1.5.0.5., Schrodinger).

**Peptide-array binding assay.** CelluSpots peptide arrays were custom synthesized and printed by Intavis, Inc. (Tübingen, Germany). Peptide sequences are shown in Supplementary Fig. 3. Arrays were blocked in 5% non-fat milk in TBST (10 mM Tris pH 8, 150 mM NaCl, 0.1% Tween-20) for 1 h at room temperature before incubation with protein solutions. His$_6$-FLAG-tagged proteins were added to a final concentration of 25 μM in 50 mM HEPES pH 7.5, 300 mM NaCl, 20% fetal bovine serum (Sigma), 0.1% NP-40, and incubated overnight at 4 °C with gentle shaking. Arrays were washed three times for 5 min with TBST, and incubated with anti-FLAG-HRP (Sigma) diluted 1/2000 in 5% non-fat milk in TBST for 1 h at room temperature. Arrays were washed three times for 5 min with TBST, and chemiluminescent signal was produced by adding SuperSignal West Femto Maximum Sensitivity Substrate (Thermo Fisher Scientific) for 5 min. Arrays were imaged on a PXi Touch imager (Syngene).

**Biolayer interferometry assay.** Biolayer interferometry binding measurements were performed with an Octet RED384 instrument (ForteBio) at 23 °C. Histone peptide binding was determined using biotinylated histone H4 1–19, either with or without acetylation of the four lysines present (S G R G **X** G G **X** G L G **X** G G A **X** R H R W G G (K-biotin); **X** = K or acetyl-K). DNA binding was performed using a 5′ biotinylated synthetic duplex 145 bp DNA (5′-CTT GCA TCG ATC CGA TTG AAC CAT CGC TCG GTG ACA GCT ACG TGA CTT AGT GTG CCC CAT CGA TCC AGT TCG ATC ACA GGC CAC CTG AGT CGA GAG TAT CGA CAC CCA GTG AAC GAT CGA TCC GAC TCG ATC GCC TCA CTG CTA G-3′) generated by scrambling the Widom-601 positioning sequence[68]. The use of 5′ biotinylated synthetic Widom-601 positioning sequence yielded identical results. Binding to nucleosome core particles was evaluated using commercially available purified human biotinylated mononucleosomes (EpiCypher). Alternatively, similar results were obtained for samples reconstituted with the EpiMark Nucleosome Assembly Kit (New England Biolabs) from 5′ biotinylated synthetic Widom-601 positioning sequence, using the dilution assembly protocol under conditions that minimize any uncomplexed DNA. Biotinylated ligands were immobilized onto Streptavidin sensors (ForteBio) and steady-state binding was determined by dipping the sensor into the indicated concentrations of purified recombinant proteins. Binding measurements were carried out in 50 mM Tris pH 7.5, 1 mM TCEP, 0.5% BSA, 0.01% Tween-20, and 50 mM NaCl (peptide) or 150 mM NaCl (DNA and nucleosome). To account for any nonspecific binding, measurements from blank sensors were subtracted for each concentration used. We note that only extremely

low levels of nonspecific binding were observed. Steady-state $K_D$ determination was performed by non-linear regression using Prism 6 (version 6.0e, GraphPad).

**Fluorescence polarization assay.** For direct binding measurements by fluorescence polarization, increasing amounts of the indicated protein were added to a 50 nM solution of a 5′ fluorescein-labeled synthetic 145 bp duplex DNA (Widom-601 sequence[68]; 5′-ATC AGA ATC CGG GTG CCG AGG CCG CTC AAT TGG TCG TAG ACA GCT CTA GCA CCG CTT AAA CGC ACG TAC GCG CTG TCC CCG CGT TTT AAC CGC CAA GGG GGA TTA CTC CCT AGT CTC CAG GCA CGT GTC AGA TAT ATA CAT CGA T-3′) in 20 mM Tris pH 7.5, 150 mM NaCl, 1% glycerol, 0.5 mg mL$^{-1}$ BSA and 0.05% Triton X-100. After 15 min incubation at room temperature, samples (20 μL) were placed in a black 384-well polypropylene plate (Greiner Bio-One) and fluorescence polarization was measured using a Wallac Victor3V 1420 Multilabel Counter (PerkinElmer). Competition assays were performed using a fixed concentration of BAZ1A-PHD (5.3 μM; apparent $K_D$) and adding increasing amounts of unlabeled synthetic duplex 145 bp DNA with identical sequence to the fluorescein-labeled probe. $K_D$ and IC$_{50}$ values and standard errors were determined by fitting the data from three independent experiments to a four-parameter binding model using Prism 6 (version 6.0e, GraphPad).

**Hydrogen-deuterium exchange mass spectrometry.** HDX-MS experiments[69–71] with uncomplexed and DNA-bound BAZ1A-PHD were performed on a fully automated Leap robotic system connected to an Orbitrap Elite mass spectrometer. Uncomplexed BAZ1A-PHD was used at a concentration of 30 μM. For the DNA-bound complex, 30 μM BAZ1A-PHD was pre-incubated with 100 μM of a 20 bp duplex DNA (5′-ACT GAC TCT GGA CTA CTG AG-3′) in phosphate-buffered saline (PBS; 10 mM PO$_4$, 137 mM NaCl, and 2.7 mM KCl, pH 7.2) prepared with H$_2$O. For deuterium ($^2$H) labeling, 3.5 μL of the sample was mixed with 55 μL of PBS buffer prepared with deuterium oxide. The samples were labeled for 30 s, 1 min, 10 min, 1 h, and 4 h at 5 °C in triplicate, and quenched with 55 μL of 8 M urea, 1 M TCEP, pH 2.2. Due to the superior digestion capability of the protease type XIII over pepsin[72], an immobilized protease XIII/pepsin (1:1 ratio) column (2.0 × 30 mm, NovaBioAssays) was used for online digestion in 0.1% formic acid and 0.04% trifluoroacetic acid in H$_2$O, flowing at 100 μL min$^{-1}$. The digests were captured on a trapping column (Waters ACQUITY BEH C$_{18}$ VanGuard Pre-column 2.1 × 5 mm), and eluted to a Waters BEH C$_{18}$ UHPLC column (2.1 × 50 mm) for peptide separation on a Waters Nano Acquity HPLC system, using a 12 min 5–50% gradient flowing at 50 μL min$^{-1}$ of buffer A (0.1% formic acid and 0.05% trifluoroacetic acid in H$_2$O) and buffer B (acetonitrile). The eluted peptides were directed onto a Thermo Orbitrap Elite mass spectrometer with electrospray ionization for detection. Mass spectra were acquired over the $m/z$ range 300–1800 at a resolving power of 60,000 at $m/z$ 400. Peptides were identified using a combination of exact mass and MS/MS aided by Mascot algorithm search (Mascot Daemon V.2.3.2; Mascot Distiller V.2.4.2). Peptide deuterium levels were determined using EXMS[73] and a script developed in-house. Complete peptide coverage was obtained for both uncomplexed and the DNA-bound BAZ1A-PHD protein, and peptides with identical sequence between the two samples (matching peptides) covered all the BAZ1A-PHD protein, except residues 32–42. The relative $^2$H uptake for peptides from uncomplexed BAZ1A-PHD is reported in Supplementary Fig. 7a as the ratio between $^2$H uptake after 4 h labeling and the theoretical maximum $^2$H uptake (without back-exchange correction), expressed as a percentage. Average relative deuterium uptake difference (ARDD; ref. [71]) was used to compute the changes in $^2$H uptake caused by DNA binding. For each HDX time point, the relative $^2$H uptake for the DNA-bound protein was subtracted to that of the uncomplexed sample, the resulting value was divided by the relative $^2$H uptake for uncomplexed BAZ1A-PHD, and expressed as a percentage. Results from all HDX time points were averaged to obtain the ARDD reported in Supplementary Fig. 7a. ARDD are reported for the six matching peptides, and could not be determined for residues 32–42.

**Transcriptional profiling.** Cells were seeded on a 100 mm plate and 20 h after seeding (~90% confluence) cells were subjected to UVC irradiation (200 J m$^{-2}$) to cause DNA damage, or subjected to a mock irradiation. 1 h after treatment or control, total RNA was extracted from ~0.75 × 10$^6$ cells using RNeasy Mini kit (QIAGEN) including on-column DNase digestion. RNA extractions were performed in triplicates. RNA concentration was determined using a NanoDrop 8000 (Thermo Scientific) and RNA integrity was determined by Fragment Analyzer (Advanced Analytical Technologies). 0.1 μg of total RNA was used as input material for library preparation using TruSeq Stranded Total RNA Library Prep kit (Illumina). Library size was confirmed using 4200 TapeStation and High Sensitivity D1K screen tape (Agilent Technologies) and concentration was determined by qPCR using Library quantification kit (KAPA). The libraries were multiplexed and then sequenced on an Illumina HiSeq4000 instrument (Illumina) to generate 30 × 10$^6$ of single-end 50 bp reads. Sequencing reads were aligned to human genome version GRCh38 with RefSeq gene models (release 67). The GSNAP alignment software (version 2013-10-10) was used to align reads to the genome with the following parameters: -M 2 -n 10 -B 2 -i 1 -N 1 -w 200000 -E 1 --pairmax-rna=200000 --clip-overlap. Gene expression levels are the sum of the number of

reads mapping to exons of RefSeq genes. Counts were normalized using size factors (as described by DESeq2) to account for both the library size and gene length. Differential expression between groups of samples was computed on the normalized counts using the R limma package's voom function. For the gene set enrichment analysis, gene sets from the Reactome database[74] through the reactome.db R package[75] were tested for overrepresentation among differentially expressed genes using a Fisher's exact test, implemented by the kegga function from the limma R package[76].

**Recombinant ACF complexes production in insect cells.** cDNAs for human BAZ1A wild-type (accession number Q9NRL2-1), BAZ1A K1181A/K1183Y and SMARCA5 (accession number O60264-1) were codon-optimized for expression in insect cells, synthesized (GenScript) and cloned into the pAcGP67A vector. BAZ1A constructs included a C-terminal His$_6$-tag, while SMARCA5 included a N-terminal FLAG-tag. cDNA-containing vectors were co-transfected with BestBac 1.0 linearized viral DNA (Expression Systems) into Sf9 cells using the TransIT transfection reagent (Mirus Bio). Resulting recombinant baculoviruses were amplified twice to produce a final viral stock suitable for protein expression. Virus quality was confirmed by gp-64 staining. For protein production, 2 L of Sf9 cells was diluted to $2 \times 10^6$ cells mL$^{-1}$ in protein-free ESF 921 media (Expression Systems) and infected with a total of 5 mL virus solution. For ACF and ACF PHD$^{KAKY}$ complexes, a BAZ1A:SMARCA5 volume ratio of 5:1 was used. Cultures were maintained at 27 °C and cells harvested 3 days post-infection by centrifugation. Cells were lysed and processed for Ni-NTA purification as described for protein produced in *E. coli*. Eluted protein was diluted 10-fold in 50 mM Tris pH 7.5, 300 mM NaCl, 1 mM TCEP, 10% glycerol, 1 mM EDTA, 0.2% Tween 20 supplemented with protease inhibitors. The samples were immediately applied to anti-FLAG M2 Affinity Gel (Sigma) for 1 h at 4 °C. After extensive washes with 50 mM Tris pH 7.5, 300 mM NaCl, 1 mM TCEP, 10% glycerol, 1 mM EDTA, 0.2% Tween 20, complexes were eluted with 150 µg mL$^{-1}$ FLAG peptide (Sigma) in 20 mM Tris pH 7.5, 200 mM NaCl, 10% glycerol, 1 mM TCEP. Purified samples were concentrated and stored as described for protein produced in *E. coli*.

**Nucleosome sliding assay.** Fluorescently labeled nucleosomes were used in nucleosome sliding assays. A Cy3-labeled DNA duplex was generated by PCR and contained 80 bp of arbitrary sequence followed by the Widom-601 nucleosome positioning sequence[68]. Nucleosomes were reconstituted with the EpiMark Nucleosome Assembly Kit (New England Biolabs). The histone octamers homogenously wrapped around the 145 bp (Widom-601 sequence) on the 3′ end of the DNA template, leaving 80 bp of free DNA protruding from the opposite end of the nucleosome core particle. Sliding reactions were carried out at 30 °C in 20 mM HEPES pH 7.9, 4.8 mM Tris, 3 mM MgCl$_2$, 12% glycerol, 0.024% IGEPAL CA-630, 30 µM FLAG peptide, 80 mM NaCl, in the absence or presence of 2 mM ATP. Reactions were carried out for 45 min using 10 nM Cy3-labeled nucleosomes, and 20 nM ACF complex or 200 nM SMARCA5. Reactions were stopped with an excess of salmon-sperm DNA (Thermo Fisher Scientific) to compete with nucleosomes for ISWI proteins, and products were separated on 6% DNA Retardation Gels (Thermo Fisher Scientific). DNA bands were visualized on a Typhoon Trio (GE Healthcare Life Sciences) using Cy3 filters.

**Data availability.** Coordinates and structure factors for the BD of BAZ1A have been deposited into the Protein Data Bank under the accession code 5UIY. The RNA sequencing data have been deposited into the Gene Expression Omnibus under the record ID GSE95227. All other data that support the findings of this study are available from the corresponding authors upon reasonable request.

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

## Acknowledgements

We thank Florence Poy and Steve F. Bellon at Constellation Pharmaceuticals, Inc. for providing initial bromodomain vectors, and Crystallographic Consulting, LLC for collecting X-ray diffraction data. We thank the Genentech peptide synthesis, cloning, expression, and next-generation sequencing groups for assistance; Yuxin Liang, Honglin Chen, and Trinna Cuellar for help with lentiviral manipulations; Asad M. Taherbhoy and Allyn Schoeffler for help with biolayer interferometry measurements; Benjamin T. Walters for help with HDX data analysis; and members of the Early Discovery Biochemistry Department, in particular Oscar W. Huang and Erin Dueber, for helpful discussion.

## Author contributions

M.O. conceived, designed, and performed most experiments and interpreted results. M.S. and M.O. collected immunofluorescence images. B.H. designed and provided constructs for genome editing. H.-M.Z. and J.Z. performed HDX experiments. S.K.K. analyzed the RNA sequencing data. J.S. solved and refined the BAZ1A bromodomain structure. E.M. F., T.B. and C.C. helped to produce protein reagents. A.G.C. designed and produced histone peptide arrays. M.O. and A.G.C. wrote the manuscript with contributions from M.S., B.H. and S.J. A.G.C. supervised the work.

## Additional information

**Competing interests:** All authors are or were employees of Genentech, Inc. The authors declare no competing financial interests.

