## [Peer Review File · Nature Communications]

Reviewers' comments:

Reviewer #1 (Remarks to the Author):

- . Summary of the key results
 - . Originality and interest: if not novel, please give references
 - . Data & methodology: validity of approach, quality of data, quality of presentation
 - . Appropriate use of statistics and treatment of uncertainties
 - . Conclusions: robustness, validity, reliability
 - . Suggested improvements: experiments, data for possible revision
 - . References: appropriate credit to previous work?
- Clarity and context: lucidity of abstract/summary, appropriateness of abstract, introduction and conclusions

Oppikofer et al. describe the functional analysis of the conserved PHD and bromodomains of BAZ1A, a subunit of the ACF chromatin remodeler complex involved in DNA damage repair. In addition to BAZ1A, ACF contains SMARCA5, the catalytic subunit. How the non-catalytic subunits, including the chromatin recognition modules, contribute to function of ISWI chromatin remodelers is poorly understood. The experiments described in this manuscript have established that BAZ1A and its paralog, BAZ1B, are involved in the DNA damage response by recruiting SMARCA5 to sites of DNA damage. In contrast to earlier reports on the mouse KO of BAZ1A, the deletion in human cell lines results in a defect in the DNA damage response. Using this phenotype as a starting point, Oppikofer et al. investigate the contribution of the chromatin reader domains of BAZ1A and BAZ1B. While the bromodomain of BAZ1A does not bind to acetylated histone peptides, that of BAZ1B does. The authors find that the PHD domains have also non-canonical substrate requirements: While the PHD domain of BAZ1A binds to DNA, that of BAZ1B does not. The BAZ1A bromo- and PHD-domains also contribute to BAZ1A function in cells: While a bromodomain gain-of-function mutant does not severely impact DNA damage repair, it nevertheless impacts recruitment of SMARCA5. In contrast, while a PHD mutant that abolishes DNA binding does impair the DNA damage response, it does not impact recruitment of SMARCA5. Thus, the phenotypes arising from both the Bromo- and PHD-domain mutants do not directly correlate with the assigned function of BAZ1A. Considering that BAZ1B, a paralog that naturally contains a functional bromodomain and a DNA binding-deficient PHD domain, would one not have expected a similar phenotype to what is seen with the BAZ1A mutants? While the provided data advances our understanding of the contribution of BAZ1A and BAZ1B to the DNA damage response, this is not uncontroversial considering the lack of a phenotype in the mouse KO of BAZ1A. The molecular details of how the reader domains contribute also remain unclear. While the data generally seems of high quality it is not clear if the overall findings are of sufficient interest to pass the threshold that would be required for publication in Nature Communications.

Major comments:

The authors report that both the PHD and bromodomain are non-canonical in their substrate requirements. However, they do not establish what the non-canonical functions of these domains are. The rescue experiment for the Bromodomain is discussed at length but considering that it is a non-physiological variant these points should be abbreviated. E.g. the discussion seems too long (4 pages) and would benefit from being tightened up.

Similarly, the analysis of the PHD domain remains somewhat unsatisfactory. While the domain can interact (weakly) with DNA in vitro and mutation of the DNA binding residues results in a variant that fails to complement BAZ1A-KO, SMARCA5 recruitment is not affected. As the authors' model is that BAZ1A helps to recruit SMARCA5 to sites of DNA damage, the physiological role of DNA binding remains unclear. As the paralog BAZ1B has a PHD domain that fails to bind DNA, it nevertheless contributes to DNA damage recovery. Thus the physiological requirement for DNA binding remains unclear. To address the issue, the authors might want to introduce the reverse mutations into BAZ1B (V1428E and A1217K/Y1219K) and test for the phenotype.

Line 883: The title of Figure legend 1 needs to be revised. The author's probably mean to say: 'The non-catalytic ISWI subunits BAZ1A and BAZ1B play a role in the response to DNA damage'

Fig. 2A: It is not clear if the BAZ1A and BAZ1B signals are above background raising the question if these proteins are really recruited in response to DNA damage. It is important to show disappearance of the signal in KO cell lines to ensure that the weak signal obtained is due to the presence of the respective proteins analyzed and not due to non-specific background signal.

The authors suggest that BAZ1A and BAZ1B 'cooperate' (line 158), are 'both required' (line 82) and 'jointly' (lines 179, line 899) recruit SMARCA5. However evidence for cooperativity is not presented. It would be more appropriate to state (eg. line 158): 'BAZ1A and BAZ1B promote recruitment of SMARCA5 to sites of DNA damage'.

Line 129: The sentence implies that the PHD module binds to DNA in an 'atypic' fashion. As the authors do not establish a DNA binding mode, it is not correct to claim it is 'atypic'. Do the authors mean to say: 'The PHD domain of BAZ1A is an unusual/atypic DNA binding domain'?

Line 898: The title needs to be revised: If it were true that BAZ1A and BAZ1B jointly recruit SMARCA5, KO of either factor would be expected to abolish SMARCA5 recruitment. The observations are however more subtle, especially for BAZ1A.

Fig4CD: Considering the DNA geometry of the nucleosome core particle, it is surprising that the BAZ1A PHD domain binds to both nucleosomes and naked DNA equally well. Can the authors comment? Considering the weak binding affinities detected, do the authors think that the detected interaction is biologically meaningful or simply arises due to charge-based interactions?

It is troubling that the BAZ1A (K1181A/K1183Y) mutant gets recruited to sites of DNA damage, indicating that DNA binding through the atypical PHD domain is not an important aspect of the recruitment process. How then do the authors explain the phenotype of this mutant as seen in Fig. 4B?

Also, what is the authors' explanation for the fact that the BD E1515V mutant only shows a mild phenotype on cell growth (Fig. 4B) but leads to 30% reduction of SMARCA5 recruitment to sites of DNA damage? Does this suggest that recruitment of SMARCA5 is not an important aspect of the DNA damage response? It is important to address these issues especially considering that cells derived from mouse *Baz1A*^{-/-} cells show no major phenotype.

Minor Comments

There are several inappropriate interjections such as "importantly", "interestingly", "significantly" etc. (Please remove and let readers decide what is remarkable about the presented findings) .

Table 1: Asterisks are missing for the values for the high-resolution shell

Reviewer #2 (Remarks to the Author):

In this manuscript, Oppikofer et al. presented a study showing that BAZ1A PHD domain interacts with DNA while BAZ1A bromodomain (BD) does not bind to acetylated lysine histone peptides, although it contains the conserved residue Asn. Both BAZ1A and BAZ1B are important for recruiting SMARCA5 to the DNA damaged chromatin site for recovery. This study provides new insights into the roles of BAZ1A and BAZ1B in the DNA recovery process. However, there are several serious concerns about the experimental design and interpretation, which directly affect

authors' conclusions. Below are the specific comments:

1, Genetic removal of BAZ1A impairs recovery after DNA damage

In this section, the authors used the BAZ1A-KO and BAZ1B-KO cells and tried to present that each of the ISWI non-catalytic subunits BAZ1A and BAZ1B plays an important role in the response of human cells to DNA damage. However, they only reintroduced the BAZ1A to BAZ1A-KO cells to examine the DNA recovery, and in later sections claimed that BAZ1B was more important for recruiting SMARCA5 to recover DNA damage. I will like to see the results of BAZ1B reintroduced to BAZ1B-KO cells to complete the overall observations.

2, BAZ1A cooperates with BAZ1B to promote the recruitment of SMARCA5 to sites of DNA damage

In this section, the authors tried to indicate that BAZ1A and BAZ1B jointly promote the recruitment of SMARCA5 to chromatin lesions. In the study, BAZ1A-KO cells showed 50% reduction of SMARCA5 enrichment, while BAZ1B-KO cells had striking 88% reduction (12% recruitment) of SMARCA5 enrichment, but I am not sure why BAZ1A-KnockOut and BAZ1B-Knockdown combination only showed 85% reduction (15% recruitment) of SMARCA5 enrichment, which is the same as the BAZ1B-KO alone recruiting, I was confused as to whether the change was from the combination effect or from the dominate effect by BAZ1B. In order to clarify the statement, the authors need to work on additional experiments such as BAZ1A-knockdown, BAZ1B-knockdown, BAZ1B-Knockout /BAZ1A-Knockdown combination, and BAZ1A-KO/BAZ1B-KO combination on the SMARCA5 enrichment so as to evaluate overall effects.

Based on the results presented, I incline to believe that BAZ1A and BAZ1B act independently to recruit ATPase SMARCA5 to chromatin lesions, while BAZ1B plays a more dominant role in the process. Because if BAZ1A and BAZ1B work separately, BAZ1A-KO can have 50% enrichment and BAZ1B-KO 12% respectively; if both BAZ1A-KO and BAZ1B-KD are in the cell lines, and BAZ1B played the separate but major role, it is still possible that the SMARCA5 drops to 15% recruitment.

3, The bromodomain of BAZ1A is non-canonical and fails to bind to acetylated histone peptides

a), In this section, the authors triumphed the first discovery of a bromodomain (BAZ1A-BD) contained conserved residue Asn but did not bind to lysine-acetylated histone because of different 'gatekeeper' residue. However, as it is well known, bromodomains even with conserved Asn generally bind to lysine-acetylated histones in a weak or strong affinity way depending on the specific flanking residues of around the acetylated-lysine. Indeed, after carefully examining the authors' Figure 3F, I found that BAZ1A-BD wild type clearly shows binding to the lysine-acetylated peptide, although about 3 to 4 fold weaker than the E1515V mutant of BAZ1A-BD. I am troubled why authors intentionally ignored their own data and jumped to the "favored" conclusions. In order to better understand the binding of BAZ1A-BD to acetylated histone lysine, I suggest that the authors work more experiments such as NMR 15N-HSQC titration (protein to peptide ratio 1:10), EMSA, Pull-down or Western Blot. For affinity validation, I also suggest that the authors run ITC titration, Florescence Polarization or SPR binding assays for BAZ1A-BD (WT and E1515V) and BAZ1B-BD.

b), In peptide array screen experiments, each experiment is an individual collection of many variables, so the general rule recommends evaluating the observed intensity of the peptides within each array only in comparison to the other peptides on the same array, but not to compare the intensity with the screening results of other copies of the same array or other arrays. However, in Figure 3E, the authors tried to compare array results of BAZ1A-BD in one array plate to BAZ1B-BD and BAZ1A BD E1515V in other plates. Clearly, the BAZ1A-BD array plate showed bleaching clean compared to other two plates, I am surprised that it did not even display the usual weak spot stains observed from other plates indicating unspecific bindings or residual colors. The blank background of BAZ1A-BD plate is quite different from that in the BAZ1A-BD E1515V plate stains, this can be contributed by many variables such as peptide composition, antibody concentration, incubation time, inhomogeneity of the array and array functionality quotient, or the possibility of

intentionally white out. As we know, the peptide array screen experiment is very difficult to distinguish weak binding from unspecific binding or false positive binding, authors can not judge the result from the peptide array screening to conclude that BAZ1A-BD did not interact with acetylated lysine. On the other hand, based on Figure 3F plot, BAZ1A-BD WT is able to bind the peptide weakly, so I think that the authors should have tried other validation experiments like NMR 15N-HSQC titration (protein to peptide ratio 1:10), EMSA, Pull-down or Western Blot.

c), In the BAZ1A-BD E1515V peptide array screen, the authors cheered to find the mutant bromodomain showing 'strong' binding to the acetylated peptide, while the biolayer interferometry assay indicated the binding rather weak ($K_d > 550 \mu\text{M}$). The spot patterns obtained from the peptide array screening can only be qualitatively or semi-quantitatively evaluated, because the signal intensity is not directly correlated to binding affinity of peptide/protein partner. Thus, the authors cannot make a 'strong' binding statement based only on peptide array screening. For better confirmation the binding affinity, I suggest that the author work on ITC titration, Florescence Polarization or SPR binding assays, and all the binding values should be compared to BAZ1B-BD binding to acetylated lysine peptides.

4, The PHD module of BAZ1A atypically binds to DNA

In this section, I suggest that the authors present a 3D structure (crystal or NMR structure) of BAZ1A PHD in complex with DNA. The binding affinity values need to be confirmed by ITC or FP experiments.

5, The BD and PHD modules support the role of BAZ1A in responding to DNA damage

In this section, I do not understand why the authors only chose to mutate E1515V in BAZ1A-BD. I guess that the authors wanted to enhance the binding ability of BAZ1A-BD to acetylated lysine, but how the interaction to acetylated lysine could contribute to BAZ1A in responding to DNA damage? What could other random mutations on BAZ1A-BD have any effect to the DNA damage? The authors did not address above questions in this section. I would suggest that the authors mutate at least five more residues around the BAZ1A-BD binding pocket to prove that E1515V is the key residue involved in the reduction (30%) in the recruitment of SMARCA5 to the DNA damaged site, especially, including mutation of the conserved Asn-1509 to totally shutdown the bromodomain's ability to bind acetylated lysine. It will also be important to consider mutation of double or triple residues or corresponding residues in BAZ1B-BD to compare to BAZ1A-BD and present more comprehensive evaluations of key residues from either BAZ1A-BD or BAZ1B-BD on the recruitment of SMARCA5.

Reviewer #3 (Remarks to the Author):

This is an important contribution with a novel finding about the role of noncatalytic subunits in a ISWI variant, which are essential for the recruitment of the ATPase subunit SMARCA5 to repair foci. The authors make mutant cells in BAZ1A (ACF1), and BAZ1B (WSTF), find a failure to recruit SMARCA5, and then use structural approaches to explain the phenomenon. I think it is an excellent and timely contribution. The field was missing this important link between SMARCA5 and recruitment modules. The structural work is compelling and thus the ms brings crucial insight to the field.

The only weaknesses are with respect to the types of damage inflicted and clarity as to what kind of damage ISWI is responding to - and whether its role is recovery or break processing. Moreover the appearance of the foci in Figure 2 is strange (see below). All other points are well covered (methodology, conclusions, references, clarity and context are appropriately explained).

1. The authors use UVC and phleomycin (antibiotic that creates ss and ds breaks) to inflict damage. The phleomycin damage and UVC damage are very different. It is odd but noteworthy

that the knockouts show a similar response to both. This needs clarification. The phleomycin damage sensitivity should be in the main figure and then some comment on type of repair or damage that recruits this remodeler is necessary.

2. Also with respect to the damage analysis: the foci in Figure 2 are very strange. Why are the foci so large ? what is the pattern ? the porous grid ? The weakness of the paper actually is the lack of clarity about the pathway(s) of repair is being elicited. The focus accumulation and disappearance should be performed under conditions that are known to promote one type of repair vs another.

Reviewers' comments and rebuttal

Reviewer #1 (Remarks to the Author):

- . Summary of the key results
 - . Originality and interest: if not novel, please give references
 - . Data & methodology: validity of approach, quality of data, quality of presentation
 - . Appropriate use of statistics and treatment of uncertainties
 - . Conclusions: robustness, validity, reliability
 - . Suggested improvements: experiments, data for possible revision
 - . References: appropriate credit to previous work?
- Clarity and context: lucidity of abstract/summary, appropriateness of abstract, introduction and conclusions

Oppikofer et al. describe the functional analysis of the conserved PHD and bromodomains of BAZ1A, a subunit of the ACF chromatin remodeler complex involved in DNA damage repair. In addition to BAZ1A, ACF contains SMARCA5, the catalytic subunit. How the non-catalytic subunits, including the chromatin recognition modules, contribute to function of ISWI chromatin remodelers is poorly understood. The experiments described in this manuscript have established that BAZ1A and its paralog, BAZ1B, are involved in the DNA damage response by recruiting SMARCA5 to sites of DNA damage. In contrast to earlier reports on the mouse KO of BAZ1A, the deletion in human cell lines results in a defect in the DNA damage response. Using this phenotype as a starting point, Oppikofer et al. investigate the contribution of the chromatin reader domains of BAZ1A and BAZ1B. While the bromodomain of BAZ1A does not bind to acetylated histone peptides, that of BAZ1B does. The authors find that the PHD domains have

also non-canonical substrate requirements: While the PHD domain of BAZ1A binds to DNA, that of BAZ1B does not. The BAZ1A bromo- and PHD-domains also contribute to BAZ1A function in cells: While a bromodomain gain-of-function mutant does not severely impact DNA damage repair, it nevertheless impacts recruitment of SMARCA5. In contrast, while a PHD mutant that abolishes DNA binding does impair the DNA damage response, it does not impact recruitment of SMARCA5. Thus, the phenotypes arising from both the Bromo- and PHD-domain mutants do not directly correlate with the assigned function of BAZ1A.

We agree with the assessment of Reviewer 1, with the exception of the conclusion stated in the last sentence above (lack of correlation with BAZ1A function). In the multiple assays we used, PHD and bromodomain (BD) mutants do not always produce phenotypes as severe as the complete *BAZ1A* knockout, but it would be surprising if they did so, given that they target single domains in a large, multi-domain protein. Nevertheless, they do produce effects of varying strength (in the same assays in which the full knockout shows defects) that are different from one another and therefore reveal aspects of the

function of each domain (and mutation) within the larger protein, a topic currently unaddressed in other studies. Perhaps the differences between the apparent relative importance of the PHD in damage recovery and SMARCA5 recruitment are what the Reviewer meant by a lack of direct correlation. We believe this to be one of the most interesting findings of the original study, as it indicates that the recruitment of SMARCA5 to sites of DNA damage is not the *only* function of BAZ1A relevant to damage recovery. In this revised manuscript, we now provide ample evidence that this is the case for both BAZ1A and BAZ1B. Indeed, mutations that inactivate the BDs of each paralog, have strong effects on DNA damage recovery yet leave SMARCA5 recruitment unaffected. These data indicate that SMARCA5 accumulation at chromatin lesions is not sufficient to promote efficient recovery from DNA damage: the regulatory factors BAZ1A and BAZ1B are also required, presumably for proper chromatin remodeling or recruitment of additional factors at lesions. We elaborate on this in our revised Discussion.

Considering that BAZ1B, a paralog that naturally contains a functional bromdomain and a DNA binding-deficient PHD domain, would one not have expected a similar phenotype to what is seen with the BAZ1A mutants?

The Reviewer is correct that these paralogs are very similar, and we do show, consistent with the Reviewer's question, that the phenotypes in *BAZ1A* and *BAZ1B* knockout lines are similar (Figures 1 and 2: slower growth after DNA damage, and defects in recruitment of SMARCA5 to sites of damage). However, BAZ1A and BAZ1B are distinct proteins. For instance, a previous study has shown that the N-terminal region of BAZ1B possesses a non-canonical tyrosine kinase activity that has not been reported for BAZ1A (Xiao et al., *Nature*, 2009). We know of no reason why all details of their biochemical and cellular functions should be identical. In fact, we show that they are not, and we believe that this increases the interest of the study.

While the provided data advances our understanding of the contribution of BAZ1A and BAZ1B to the DNA damage response, this is not uncontroversial considering the lack of a phenotype in the mouse KO of BAZ1A. The molecular details of how the reader domains contribute also remain unclear. While the data generally seems of high quality it is not clear if the overall findings are of sufficient interest to pass the threshold that would be required for publication in *Nature Communications*.

We do not understand fully this comment of the Reviewer. It would appear that the Reviewer considers the experimental results of high quality, an advance in the field, and potentially provocative or controversial, yet is not sure that these attributes render the work interesting. Moreover in this revised version of the manuscript, we provide a large amount of additional data, that increase the interest of the work (e.g., we expanded our study of BAZ1B), and include a more detailed investigation of the DNA damage hypersensitivity caused by BAZ1A-PHD function-altering mutations.

Major comments:

The authors report that both the PHD and bromodomain are non-canonical in their substrate requirements. However, they do not establish what the non-canonical functions of these domains are. The rescue experiment for the Bromodomain is discussed at length but considering that it is a non-physiological variant these points should be abbreviated. E.g. the discussion seems too long (4 pages) and would benefit from being tightened up.

We edited our discussion, and focused our attention on observations that are directly relevant to the physiological functions of BAZ1A (and BAZ1B).

Similarly, the analysis of the PHD domain remains somewhat unsatisfactory. While the domain can interact (weakly) with DNA in vitro and mutation of the DNA binding residues results in variant that fails to complement BAZ1A-KO, SMARCA5 recruitment is not affected. As the authors' model is that BAZ1A helps to recruit SMARCA5 to sites of DNA damage, the physiological role of DNA binding remains unclear. As the paralog BAZ1B has a PHD domain that fails to bind DNA, it nevertheless contributes to DNA damage recovery. Thus the physiological requirement for DNA binding remains unclear. To address the issue, the authors might want to introduce the reverse mutations into BAZ1B (V1428E and A1217K/Y1219K) and test for the phenotype.

We reach the same conclusion as the Reviewer: the PHD-DNA interaction discovered here is dispensable for recruitment of SMARCA5 at sites of DNA lesions, and yet critical for surviving DNA damage. As noted above, SMARCA5 recruitment is likely not the *only* function of BAZ1A, and we consider this to be a surprising and significant finding. This observation is sensible, given that, as noted by the Reviewer, BAZ1B is critical for SMARCA5 recruitment (even more so than BAZ1A) and yet its PHD module fails to bind DNA.

As correctly noted by the Reviewer, the BAZ1A PHD-DNA interaction discovered here is relatively weak ($K_D \sim 5 \mu\text{M}$), and there is no reason to believe that this *general* DNA interaction would be required for *specific* recruitment to sites of DNA damage. The additional data provided in this revised manuscript (GFP-BAZ1A localization at sites of damage, and transcriptional profiling) strongly suggest that BAZ1A-PHD has complex, critical functions beyond SMARCA5 recruitment. Moreover, we now provide multiple examples of cell lines expressing mutant BAZ1A or BAZ1B that support SMARCA5 recruitment but are nevertheless hypersensitive to DNA damage. These data show clearly that, while critical, SMARCA5 recruitment is not the only function of BAZ1A and BAZ1B.

We agree with the Reviewer that an assessment of BAZ1B beyond what we had included in the original study is interesting and that the phenotypes of BAZ1B mutants could clarify the function of this paralog in responding to DNA damage. The BAZ1B-PHD A1217K/Y1219K mutant does not bind DNA in the biophysical assays used for BAZ1A-PHD (Figure 6), so we did not evaluate this mutant in phenotypic damage assays. However, we successfully generated clonal cell lines that re-express BAZ1B V1425E (gatekeeper mutant) and BAZ1B N1419Y (asparagine anchor mutant). Both mutations caused DNA damage hypersensitivity, showing that acetyl-lysine recognition by BAZ1B-BD is important for recovery from DNA damage.

Line 883: The title of Figure legend 1 needs to be revised. The author's probably mean to say: 'The non-catalytic ISWI subunits BAZ1A and BAZ1B play a role in the response to DNA damage'

We revised the title.

Fig. 2A: It is not clear if the BAZ1A and BAZ1B signals are above background raising the question if these proteins are really recruited in response to DNA damage. It is important to show disappearance of the signal in KO cell lines to ensure that the weak signal obtained is due to the presence of the respective proteins analyzed and not due to non-specific background signal.

We agree with the Reviewer that the proposed control is useful and increases confidence in the reported result. Additional laser microirradiation experiments in *BAZ1A*- and *BAZ1B*-knockout cells are now included in Supplementary Fig. 2B. We observe no apparent recruitment of BAZ1A or BAZ1B in the respective knockout cells, consistent with our original assignment of the antibody staining as specific.

The authors suggest that BAZ1A and BAZ1B 'cooperate' (line 158), are 'both required' (line 82) and 'jointly' (lines 179, line 899) recruit SMARCA5. However evidence for cooperativity is not presented. It would be more appropriate to state (eg. line 158): 'BAZ1A and BAZ1B promote recruitment of SMARCA5 to sites of DNA damage'.

We agree with the Reviewer that the original wording might have been misleading and have revised the text to clarify this point. We did not intend to imply cooperativity in the molecular sense.

Line 129: The sentence implies that the PHD module binds to DNA in an 'atypical' fashion. As the authors do not establish a DNA binding mode, it is not correct to claim it is 'atypical'. Do the authors mean to say: 'The PHD domain of BAZ1A is an unusual/atypical DNA binding domain'?

Yes, we meant to say that the PHD module is a non-canonical DNA binding domain and that this function is unusual among PHD modules, without any regard to the detailed binding mode (currently unknown). We revised the text to clarify this point.

Line 898: The title needs to be revised: If it were true that BAZ1A and BAZ1B jointly recruit SMARCA5, KO of either factor would be expected to abolish SMARCA5 recruitment. The observations are however more subtle, especially for BAZ1A.

We agree with the Reviewer and revised the title accordingly.

Fig4CD: Considering the DNA geometry of the nucleosome core particle, it is surprising that the BAZ1A PHD domain binds to both nucleosomes and naked DNA equally well. Can the authors comment? Considering the weak binding affinities detected, do the

authors think that the detected interaction is biologically meaningful or simply arises due to charge-based interactions?

We were initially surprised by this as well, but the size of BAZ1A-PHD is arguably small enough to cover only a small footprint, such that the curvature of DNA around the nucleosome might not significantly impact its binding affinity. We now map the PHD-DNA binding surface (by HDX-MS) as likely to be located toward the C-terminal region of BAZ1A-PHD. This region appears to be rather flexible and may adapt to the geometry of the DNA around the nucleosome. Furthermore, given the relative sizes of BAZ1A-PHD and the DNAs, it is likely that multiple copies of BAZ1A-PHD bind simultaneously. However, these multiple binding events appear to be independent with similar K_D s (reading out as an average value). This averaging may obscure any subtle differences that could exist in binding at individual sites along the DNA in the nucleosome or nucleosome-free contexts.

The surface electrostatic properties of BAZ1A-PHD and BAZ1B-PHD are very similar, with the exception of the two lysines that are critical for DNA binding by BAZ1A-PHD. Nevertheless, the introduction of these solvent-exposed lysines to the BAZ1B-PHD module (A1217K/Y1219K) does not enable this domain to bind DNA. This demonstrates that the mere presence of positive charges is not sufficient to drive DNA binding and suggests that BAZ1A-PHD binding to DNA is specific and therefore likely to be meaningful. In any case, the mutant data shown in Figure 7B argue strongly for biological relevance without ruling out interactions with molecules other than DNA that require the same basic residues of BAZ1A.

It is troubling that the BAZ1A (K1181A/K1183Y) mutant gets recruited to sites of DNA damage, indicating that DNA binding through the atypical PHD domain is not an important aspect of the recruitment process. How then do the authors explain the phenotype of this mutant as seen in Fig. 4B?

It is true that the mutation of BAZ1A-PHD that impairs DNA binding also impairs recovery from DNA damage without appreciably affecting the recruitment of SMARCA5 at sites of damage. However, we do not interpret this as “troubling” because we would not necessarily expect an ubiquitous factor such as DNA itself to be the recruitment signal to DNA damage. Instead, we hypothesize that the PHD-DNA interaction discovered here might help to orient or to regulate the catalytic output of the BAZ1A-SMARCA5 complex. While we did not observe a clear reduction in the repositioning activity of a BAZ1A-SMARCA5 complex bearing a mutant PHD module (ACF^{KAKY}) when using a reconstituted mono-nucleosomal substrate, it remains possible that this non-canonical PHD module plays an important role in the remodeling of chromatin in physiological settings. Moreover, we now provide evidence that in addition to a potential role at sites of damaged DNA, BAZ1A and a functional PHD module are required for maintaining the gene expression profiles seen for wild-type cells before and after DNA damage.

Also, what is the authors' explanation for the fact that the BD E1515V mutant only shows

a mild phenotype on cell growth (Fig. 4B) but leads to 30% reduction of SMARCA5 recruitment to sites of DNA damage? Does this suggest that recruitment of SMARCA5 is not an important aspect of the DNA damage response? It is important to address these issues especially considering that cells derived from mouse *Baz1A*^{-/-} cells show no major phenotype.

While the hypersensitivity of *BAZ1A* E1515V is mild, it is nevertheless consistent across three independent clonal lines (see revised Fig. 4 and Supplementary Fig. S4), and this is consistent with the moderate defect in SMARCA5 recruitment (relative to *BAZ1A*-knockout). The two assays are completely different and each measures the aggregate effect of a complex process. We would therefore not expect *quantitative* correlation of subtle differences when comparing the two. For example, the *BAZ1A* and *BAZ1B* knockout lines show similar defects in the growth assay (Fig. 1), while the *BAZ1B* knockout more strongly affects SMARCA5 recruitment (Fig. 2D). This might suggest that once SMARCA5 at damage foci declines below a threshold, cell growth is impacted. Overall, we believe that SMARCA5 recruitment is important for recovery from DNA damage but not sufficient (based on the BD and PHD mutant data). Our data do not directly address the reason for the lack of observed damage phenotypes in the mouse knockout studies, although we do acknowledge the difference and provide some possible explanations in our Discussion.

Minor Comments

There are several inappropriate interjections such as "importantly", "interestingly", "significantly" etc. (Please remove and let readers decide what is remarkable about the presented findings) .

We acknowledge the Reviewer's different preference in writing style; however, many readers find such "interjections" to be useful transitions that improve readability. We have reviewed such usage and removed those where clarity was not adversely impacted.

Table 1: Asterisks are missing for the values for the high-resolution shell

We corrected this.

Reviewer #2 (Remarks to the Author):

In this manuscript, Oppikofer et al. presented a study showing that *BAZ1A* PHD domain interacts with DNA while *BAZ1A* bromodomain (BD) does not bind to acetylated lysine histone peptides, although it contains the conserved residue Asn. Both *BAZ1A* and *BAZ1B* are important for recruiting SMARCA5 to the DNA damaged chromatin site for recovery. This study provides new insights into the roles of *BAZ1A* and *BAZ1B* in the DNA recovery process. However, there are several serious concerns about the experimental design and interpretation, which directly affect authors' conclusions. Below are the specific comments:

1, Genetic removal of BAZ1A impairs recovery after DNA damage

In this section, the authors used the BAZ1A-KO and BAZ1B-KO cells and tried to present that each of the ISWI non-catalytic subunits BAZ1A and BAZ1B plays an important role in the response of human cells to DNA damage. However, they only reintroduced the BAZ1A to BAZ1A-KO cells to examine the DNA recovery, and in later sections claimed that BAZ1B was more important for recruiting SMARCA5 to recover DNA damage. I will like to see the results of BAZ1B reintroduced to BAZ1B-KO cells to complete the overall observations.

Our original focus was on the role of BAZ1A, but we agree that the suggested experiment is useful. We genome-edited and cloned a *BAZ1B*-knockout cell line that re-expresses BAZ1B and showed that this rescues the DNA damage hypersensitivity of *BAZ1B*-knockout cells.

2, BAZ1A cooperates with BAZ1B to promote the recruitment of SMARCA5 to sites of DNA damage

In this section, the authors tried to indicate that BAZ1A and BAZ1B jointly promote the recruitment of SMARCA5 to chromatin lesions. In the study, BAZ1A-KO cells showed 50% reduction of SMARCA5 enrichment, while BAZ1B-KO cells had striking 88% reduction (12% recruitment) of SMARCA5 enrichment, but I am not sure why BAZ1A-KnockOut and BAZ1B-Knockdown combination only showed 85% reduction (15% recruitment) of SMARCA5 enrichment, which is the same as the BAZ1B-KO alone recruiting, I was confused as to whether the change was from the combination effect or from the dominate effect by BAZ1B. In order to clarify the statement, the authors need to work on additional experiments such as BAZ1A-knockdown, BAZ1B-knockdown, BAZ1B-Knockout /BAZ1A-Knockdown combination, and BAZ1A-KO/BAZ1B-KO combination on the SMARCA5 enrichment so as to evaluate overall effects. Based on the results presented, I incline to believe that BAZ1A and BAZ1B act independently to recruit ATPase SMARCA5 to chromatin lesions, while BAZ1B plays a more dominant role in the process. Because if BAZ1A and BAZ1B work separately, BAZ1A-KO can have 50% enrichment and BAZ1B-KO 12% respectively; if both BAZ1A-KO and BAZ1B-KD are in the cell lines, and BAZ1B played the separate but major role, it is still possible that the SMARCA5 drops to 15% recruitment.

We direct the Reviewer's attention to the similar comment from Reviewer #1 regarding use of the words "jointly" and "cooperate" and to our response. We have revised the text to state more clearly what was meant, namely that BAZ1A and BAZ1B each recruit SMARCA5 to sites of damage and that they are not redundant. We did not intend to imply cooperativity in the molecular sense.

3, The bromodomain of BAZ1A is non-canonical and fails to bind to acetylated histone peptides

a), In this section, the authors triumphed the first discovery of a bromodomain (BAZ1A-BD) contained conserved residue Asn but did not bind to lysine-acetylated histone because of different 'gatekeeper' residue. However, as it is well known, bromodomains

even with conserved Asn generally bind to lysine-acetylated histones in a weak or strong affinity way depending on the specific flanking residues of around the acetylated-lysine. Indeed, after carefully examining the authors' Figure 3F, I found that BAZ1A-BD wild type clearly shows binding to the lysine-acetylated peptide, although about 3 to 4 fold weaker than the E1515V mutant of BAZ1A-BD. I am troubled why authors intentionally ignored their own data and jumped to the "favored" conclusions. In order to better understand the binding of BAZ1A-BD to acetylated histone lysine, I suggest that the authors work more experiments such as NMR 15N-HSQC titration (protein to peptide ratio 1:10), EMSA, Pull-down or Western Blot.

For affinity validation, I also suggest that the authors run ITC titration, Florescence Polarization or SPR binding assays for BAZ1A-BD (WT and E1515V) and BAZ1B-BD.

It appears to us that the Reviewer is missing the key point, in that the mutant E1515V unquestionably binds better (and specifically) to acetylated peptides and that this increased binding leads to an effect in a biological assay. Also, the Reviewer appears to be misinterpreting the data originally shown in Figure 3F by equating a signal ratio at a given analyte concentration with a K_D ratio. The correct way to make such a comparison is to determine the shift in the midpoints of the binding curves. In the present case, this is impossible to do because there is insufficient binding by the WT protein to fit the data. One might very crudely estimate the difference by comparing concentrations of analyte at which one sees similar signal. Such a crude comparison suggests an 8–10-fold difference in affinity for the peptide used, but this isn't a valid quantitation of the K_D difference. We do not agree with the Reviewer that there is any utility in conducting additional non-quantitative experiments such as pull-down/Western (subject to avidity effects and non-linear detection issues that may lead to erroneous conclusions) or EMSA (poor sensitivity to low-level binding). Nor do we agree that FP would provide any useful information in the concentration regime under discussion ($K_D \geq 5$ mM). ITC is also unsuitable for any meaningful quantitation of WT bromodomain binding to peptide because of limitations in achievable protein concentration. SPR is a very similar method to biolayer interferometry (as used in the experiment shown in Figure 3F) and would not be expected to add any additional useful information.

Instead of focusing on estimating a K_D for the extremely weak interaction between BAZ1A-BD WT and acetylated histones, we asked whether the interaction is actually mediated by the canonical BD pocket (rather than by a non-specific interaction). To this end, we cloned, expressed and purified two additional BAZ1A-BD variants: a substitution that abrogates acetyl-lysine binding (N1509Y; BAZ1A^{NY}), and the double mutant (E1515V/N1509Y; BAZ1A^{EVNY}). Using these new tools we were able to show that the extremely weak binding of BAZ1A-BD WT requires the canonical asparagine anchor and measurably exceeds the background binding to unmodified peptides. Also, we show that the E1515V mutation promotes acetyl-lysine binding through the canonical BD pocket. We include these data in this revised version of the manuscript (Fig. 3F).

b), In peptide array screen experiments, each experiment is an individual collection of many variables, so the general rule recommends evaluating the observed intensity of the peptides within each array only in comparison to the other peptides on the same array, but

not to compare the intensity with the screening results of other copies of the same array or other arrays.

However, in Figure 3E, the authors tried to compare array results of BAZ1A-BD in one array plate to BAZ1B-BD and BAZ1A BD E1515V in other plates. Clearly, the BAZ1A-BD array plate showed bleaching clean compared to other two plates, I am surprised that it did not even display the usual weak spot stains observed from other plates indicating unspecific bindings or residual colors. The blank background of BAZ1A-BD plate is quite different from that in the BAZ1A-BD E1515V plate stains, this can be contributed by many variables such as peptide composition, antibody concentration, incubation time, inhomogeneity of the array and array functionality quotient, or the possibility of intentionally white out.

We agree with the Reviewer that peptide array screens are qualitative, not quantitative in nature. This is why we complemented our array analysis with biolayer interferometry, as described in the manuscript. Nevertheless, one may make qualitative comparisons from different arrays screened and imaged in the same manner, much as one might use western blots from two different gels in the course of the same study. The array experiments reported here were performed in parallel, using the same buffers, antibody preparations and incubation times, as described in Materials and Methods. Moreover, these results are highly reproducible across multiple arrays, including commercially available ones. We find the Reviewer's suggestion that the clean background of the BAZ1A-BD array may have been generated by "intentional white out" disturbing and unwarranted, and contrary to the Reviewer's perception, areas of the E1515V array have background signal as low as that of the WT array. BAZ1A-BD is a very soluble and well-behaved protein domain (indeed we could crystallize it), and it simply does not bind to the surface of the array in our experiments. In contrast, BAZ1B-BD is not very soluble and tends to generate a higher background. We have done hundreds of such experiments in our laboratory on dozens of bromodomains and mutants, and this kind of background signal variation is commonly seen. Finally, we believe that the original coloring superposed onto the images to delineate H4- from H3-derived peptides might have been a source of confusion, and we have removed it in the revised manuscript.

As we know, the peptide array screen experiment is very difficult to distinguish weak binding from unspecific binding or false positive binding, authors can not judge the result from the peptide array screening to conclude that BAZ1A-BD did not interact with acetylated lysine.

We don't understand this comment. The Reviewer says that it is "very difficult to distinguish weak binding from unspecific binding or false positive binding". However, we do not see binding at all on the peptide array. As stated above, we performed additional biolayer interferometry measurements to better characterize the binding of BAZ1A-BD WT to acetylated peptides.

On the other hand, based on Figure 3F plot, BAZ1A-BD WT is able to bind the peptide weakly, so I think that the authors should have tried other validation experiments like

NMR 15N-HSQC titration (protein to peptide ratio 1:10), EMSA, Pull-down or Western Blot.

The Reviewer already made this comment above (subsection a) and we addressed it there.

c), In the BAZ1A-BD E1515V peptide array screen, the authors cheered to find the mutant bromodomain showing ‘strong’ binding to the acetylated peptide, while the biolayer interferometry assay indicated the binding rather weak ($K_d > 550 \mu\text{M}$). The spot patterns obtained from the peptide array screening can only be qualitatively or semi-quantitatively evaluated, because the signal intensity is not directly correlated to binding affinity of peptide/protein partner. Thus, the authors cannot make a ‘strong’ binding statement based only on peptide array screening.

We agree with the Reviewer that, although the binding is certainly stronger than that seen for WT bromodomain, the binding is not “strong” in an absolute sense. We revised the text accordingly.

For better confirmation the binding affinity, I suggest that the author work on ITC titration, Florescence Polarization or SPR binding assays, and all the binding values should be compared to BAZ1B-BD binding to acetylated lysine peptides.

Again, the Reviewer already made this comment about additional methodologies, and we addressed it above. Also, we reiterate that BAZ1B-BD is significantly less soluble than BAZ1A-BD and that, accordingly, the scope of feasible biophysical studies is considerably limited.

4, The PHD module of BAZ1A atypically binds to DNA

In this section, I suggest that the authors present a 3D structure (crystal or NMR structure) of BAZ1A PHD in complex with DNA. The binding affinity values need to be confirmed by ITC or FP experiments.

As discussed in our response to Reviewer #1, the word “atypically” might have generated some confusion. We did not mean that the molecular details of the interaction of BAZ1A-PHD and DNA were atypical, but instead that PHD modules typically do not bind DNA. We therefore revised the wording for clarity. Because we did not intend to comment on the binding mode, we believe that determining a 3D structure for BAZ1A-PHD in complex with DNA is both unnecessary and beyond the scope of this work.

To further validate the interaction of BAZ1A-PHD and DNA we employed an orthogonal method. As suggested by the Reviewer we used fluorescence polarization (FP), in both direct and competition formats, and found that in solution BAZ1A-PHD readily binds DNA ($K_D = 5 \pm 1 \mu\text{M}$), in a fashion dependent on the K1181 and K1183 residues. Solution FP measurements are a useful complement to biolayer interferometry, where the analyte is bound to a surface and binding might consequently be affected. These data are now included in Supp. Fig. S6C,D. Moreover, we used hydrogen-deuterium exchange mass spectrometry (HDX-MS; Fig. 6D,E and Supp. Fig. S7A–C) to map the DNA binding surface of BAZ1A-PHD. Comparing the deuterium uptake of uncomplexed and

DNA-bound BAZ1A-PHD revealed that DNA binding occurs in solution and likely involves residues toward the C-terminus of BAZ1A-PHD, encompassing K1181 and K1183, as expected from our mutagenesis studies.

5, The BD and PHD modules support the role of BAZ1A in responding to DNA damage. In this section, I do not understand why the authors only chose to mutate E1515V in BAZ1A-BD. I guess that the authors wanted to enhance the binding ability of BAZ1A-BD to acetylated lysine, but how the interaction to acetylated lysine could contribute to BAZ1A in responding to DNA damage?

This is correct, the aim of the E1515V mutation is to promote binding of BAZ1A-BD to acetyl-lysine containing species. Our goal was not necessarily to improve the response to DNA damage, but to test (agnostically) whether a gain-of-function bromodomain mutant would affect the overall function of BAZ1A in cells. Our data indicate that indeed the E1515V mutation impairs the cellular function of BAZ1A, possibly by engaging unproductive acetyl-lysine containing binding partners.

What could other random mutations on BAZ1A-BD have any effect to the DNA damage? The authors did not address above questions in this section. I would suggest that the authors mutate at least five more residues around the BAZ1A-BD binding pocket to prove that E1515V is the key residue involved in the reduction (30%) in the recruitment of SMARCA5 to the DNA damaged site, especially, including mutation of the conserved Asn-1509 to totally shutdown the bromodomain's ability to bind acetylated lysine. It will also be important to consider mutation of double or triple residues or corresponding residues in BAZ1B-BD to compare to BAZ1A-BD and present more comprehensive evaluations of key residues from either BAZ1A-BD or BAZ1B-BD on the recruitment of SMARCA5.

The generation of genome-edited clonal cell lines is very laborious, as is their functional characterization. Therefore, the study of "at least five more" "random mutations" by these techniques does not appear to us to be an appropriate experimental approach. We do not see how an investigation of BAZ1B BD mutants would provide any meaningful insight into the effect of the BAZ1A E1515V mutant, nor do we understand what hypothesis the Reviewer is suggesting we address through double or triple mutagenesis of BAZ1A-BD. However, we believe that asparagine anchor mutations for BAZ1A and BAZ1B are sensible experiments that can inform on the potential cellular function of their bromodomains. Therefore, we generated 7 additional genome-edited, clonal cell lines, in addition to the 5 lines originally included in the manuscript. We generated mutant lines bearing the N1509Y mutation (to abolish acetyl-lysine binding) in the context of WT and E1515V BAZ1A, or lacking the BAZ1A-BD altogether (BAZ1A^{ΔBD}). In addition we generated mutant BAZ1B cell lines bearing the N1419Y or the V1425E mutations (both expected to decrease or abolish acetyl-lysine binding). We characterized these cell lines for SMARCA5 accumulation at sites of DNA damage as well as proliferation before and after a DNA damaging event. These data are now provided in Figures 4, S4, 5, and S5, and the results are discussed in the revised manuscript. In

summary, while all mutations produce some level of DNA damage hypersensitivity, only the BAZ1A E1515V mutation causes a moderate reduction in SMARCA5 recruitment to DNA lesions. Possible explanations for this observation are included in the Discussion.

Reviewer #3 (Remarks to the Author):

This is an important contribution with a novel finding about the role of noncatalytic subunits in a ISWI variant, which are essential for the recruitment of the ATPase subunit SMARCA5 to repair foci. The authors make mutant cells in BAZ1A (ACF1), and BAZ1B (WSTF), find a failure to recruit SMARCA5, and then use structural approaches to explain the phenomenon. I think it is an excellent and timely contribution. The field was missing this important link between SMARCA5 and recruitment modules. The structural work is compelling and thus the ms brings crucial insight to the field.

The only weaknesses are with respect to the types of damage inflicted and clarity as to what kind of damage ISWI is responding to - and whether its role is recovery or break processing. Moreover the appearance of the foci in Figure 2 is strange (see below). All other points are well covered (methodology, conclusions, references, clarity and context are appropriately explained).

We thank the Reviewer for recognizing the quality and impact of our work. As described in the introduction section of the manuscript, BAZ1A, SMARCA5, and other ISWI-associated factors, have been shown to respond to multiple types of DNA damage. We agree that it is important to clarify what type of DNA damage is produced in the various experimental approaches we used. See specific additions and revisions below.

1. The authors use UVC and phleomycin (antibiotic that creates ss and ds breaks) to inflict damage. The phleomycin damage and UVC damage are very different. It is odd but noteworthy that the knockouts show a similar response to both. This needs clarification. The phleomycin damage sensitivity should be in the main figure and then some comment on type of repair or damage that recruits this remodeler is necessary.

We agree with the Reviewer, and we therefore moved the data generated with phleomycin D1 treatment to the main figure. Published work shows that BAZ1A and BAZ1B are important for multiple types of DNA damage repair, including nucleotide excision repair, non-homologous end joining and homologous recombination. Therefore, it is likely that BAZ1A and BAZ1B, as well as SMARCA5, play a general and fundamental role in DNA damage repair. One possibility is that these ISWI factors remodel chromatin at sites of DNA damage to increase DNA accessibility for repair. Alternatively, they may promote the reassembly of chromatin after DNA repair has been completed. In any case, the molecular interactions uncovered here suggest some ways forward to define the exact role of ISWI remodelers in responding to DNA damage. We revised the manuscript to include these thoughts.

2. Also with respect to the damage analysis: the foci in Figure 2 are very strange. Why are the foci so large ? what is the pattern ? the porous grid ? The weakness of the paper actually is the lack of clarity about the pathway(s) of repair is being elicited. The focus accumulation and disappearance should be performed under conditions that are known to promote one type of repair vs another.

The size of the γ -H2AX induced by UVC irradiation is directly linked to the 5 μ m porous membrane used for these experiments. The size and shape of the foci produced here are fully consistent with that of foci reported previously using the same method (Suzuki et al., Nature Protocols, 2011).

Very high doses of UVC irradiation alone can cause the appearance of some γ -H2AX staining coupled to the repair of helix-distorting DNA lesions by nucleotide excision repair. However, the appearance of γ -H2AX foci is best correlated with the processing of DNA double-strand breaks (DSBs). To clarify, in our experiments cells were pre-incubated with BrdU before being subjected to UVC irradiation through the 5 μ m porous membranes. Incorporation of BrdU in combination with moderate UVC irradiation causes photochemical events leading to DSBs. Consistent with the method published by Suzuki, et al., UVC irradiation alone (i.e. in absence of BrdU) produces only few and diffuse γ -H2AX foci. This indicates that the strong γ -H2AX foci reported here are primarily caused by the production of BrdU-dependent DSBs. We revised the manuscript to emphasize this point.

REVIEWERS' COMMENTS:

Reviewer #1 (Remarks to the Author):

The authors have addressed all the raised points and I feel that the paper is suitable for publication.

Reviewer #2 (Remarks to the Author):

In the revised manuscript, Oppikofer et al. clarified the role of BAZ1A or BAZ1B in DNA damage recovery as well as SMARCA5 recruitment to chromatin using genetic editing methods. They further profiled the gene expression variation in BAZ1A knockout cells and cells reconstituted with ectopic BAZ1A and BAZ1A mutants that carried with DNA binding deficient PHD domain at transcriptional level. In addition, the authors modified wording of their conclusion of some of their experiments that did not have strong evidence. Overall, this study still appears to be fractionated with individual pieces that explore the functions of the bromodomain and PHD domain in DNA damage response and SMARCA5 recruitment to the chromatin, and clearly lacked connection between different parts. Despite a lot of data presented, the authors did not provide a clear molecular mechanism. Therefore, the current study did not provide sufficient new knowledge in our understanding of protein module mediated DNA repair and gene transcription, and did not satisfy the expectation for publication in Nature communications yet. Specific concerns are listed below.

1. The authors claimed that the acetyllysine binding affinity variation of BAZ1A and BAZ1B bromodomains was caused by the different gate keeper residues in bromodomains of BAZ1A and BAZ1B and that bromodomain-acetyllysine recognition was critical for their role in DNA damage response. The claim raised the question on why BAZ1A and BAZ1B play similar functions in DNA damage response given the significant difference of bromodomain-acetyllysine binding affinity of BAZ1A and BAZ1B.
2. Are BAZ1A and BAZ1B co-expressed in cells? Do BAZ1A and BAZ1B work separately or corporately in DNA damage response or other cellular processes?
3. Base on the author's data, BAZ1B appears to be more important than BAZ1A in bromodomain-acetyllysine recognition mediated DNA damage response. I was disappointed that the authors did not offer any explanation why BAZ1B is critical to DNA binding recovery. Contrary to BAZ1A, authors indicated BAZ1B binds acetyllysine peptide but PHD domain does not interact with DNA, but I did not find any information about BAZ1B functioning in DNA damage recovery, then why BAZ1B is so important in SMARCA5 recruitment and DNA damage recovery? The authors should make a focus on BAZ1B.
4. How did BAZ1A (or BAZ1B) recruit SMAC5 to the chromatin? In other words, what is the molecular mechanism?
5. How did Bromodomain-acetyllysine recognition and PHD domain-DNA binding in BAZ1A and BAZ1B interplay with each other in the response of DNA damage?
6. The authors did not use other methods to validate BAZ1A bromodomain binding to acetyllysine weakly. This is critical for the whole study since the authors assumed bromodomain-acetyllysine recognition plays an imperative role in BAZ1A modulated DNA damage response.
7. Biolayer interferometry is not an ideal method to monitor weak protein-peptide interaction, especially, when K_d is greater than 100M. Authors needed to use other methods to validate K_d value of BAZ1A-BDEV binding to acetylated peptides or histones.
8. Results from peptide array must be validated by other biochemical (pulldown and western blot) or biophysical methods such as NMR etc.

Reviewer #3 (Remarks to the Author):

The authors have very adequately responded to the reviewers' comments and I recommend acceptance in its current state.

Reviewers' comments and rebuttal

Reviewer #2 (Remarks to the Author):

In the revised manuscript, Oppikofer et al. clarified the role of BAZ1A or BAZ1B in DNA damage recovery as well as SMARCA5 recruitment to chromatin using genetic editing methods. They further profiled the gene expression variation in BAZ1A knockout cells and cells reconstituted with ectopic BAZ1A and BAZ1A mutants that carried with DNA binding deficient PHD domain at transcriptional level. In addition, the authors modified wording of their conclusion of some of their experiments that did not have strong evidence. Overall, this study still appears to be fractionated with individual pieces that explore the functions of the bromodomain and PHD domain in DNA damage response and SMARCA5 recruitment to the chromatin, and clearly lacked connection between different parts. Despite a lot of data presented, the authors did not provide a clear molecular mechanism. Therefore, the current study did not provide sufficient new knowledge in our understanding of protein module mediated DNA repair and gene transcription, and did not satisfy the expectation for publication in Nature communications yet. Specific concerns are listed below.

We agree that our present understanding of BAZ1A and BAZ1B does not include complete molecular mechanisms that fully explain their roles in DNA damage recovery. However, we do believe that our study reveals significant new information that will be important in developing such mechanisms. We regret that Reviewer #2 does not value aspects of the study that appealed to other Reviewers, and we continue to disagree with numerous aspects of this Reviewer's critiques, as outlined below.

1. The authors claimed that the acetyllysine binding affinity variation of BAZ1A and BAZ1B bromodomains was caused by the different gate keeper residues in bromodomains of BAZ1A and BAZ1B and that bromodomain-acetyllysine recognition was critical for their role in DNA damage response. The claim raised the question on why BAZ1A and BAZ1B play similar functions in DNA damage response given the significant difference of bromodomain-acetyllysine binding affinity of BAZ1A and BAZ1B.

We are not prepared to conclude, as the Reviewer does here, that BAZ1A and BAZ1B "play similar functions in DNA damage response", nor did we make any such claim in the manuscript. The data we present demonstrate that each protein plays a role in damage recovery and that details of their functions are not the same.

2. Are BAZ1A and BAZ1B co-expressed in cells? Do BAZ1A and BAZ1B work separately or corporately in DNA damage response or other cellular processes?

The Western blot data we present (Figure 1a) show that BAZ1A and BAZ1B are co-expressed in the parental cell line we used. We addressed the second question regarding cooperativity in our revised manuscript and in the responses both to Reviewer #1 and this Reviewer in our first rebuttal letter. As we stated there, the evidence we present is consistent with independent roles for BAZ1A and BAZ1B in DNA damage recovery, and we had removed all language from the manuscript that could be interpreted as suggesting that they function "corporately [*sic*]".

3. Based on the author's data, BAZ1B appears to be more important than BAZ1A in bromodomain-acetyllysine recognition mediated DNA damage response. I was disappointed that the authors did not offer any explanation why BAZ1B is critical to DNA binding recovery. Contrary to BAZ1A, authors indicated BAZ1B binds acetyllysine peptide but PHD domain does not interact with DNA, but I did not find any information about BAZ1B functioning in DNA damage recovery, then why BAZ1B is so important in SMARCA5 recruitment and DNA damage recovery? The authors should make a focus on BAZ1B.

We disagree with the Reviewer's statement that "BAZ1B appears to be more important than BAZ1A in bromodomain-acetyllysine recognition mediated DNA damage response". Comparing Figures 4b,d (reconstitution of *BAZ1A*-KO cells with BAZ1A Δ BD or NY mutants) to Figure 5c (reconstitution of *BAZ1B*-KO cells with BAZ1B NY mutant), one would conclude that there is an important role for each bromodomain in DNA damage recovery that is highly likely to involve ligand recognition (disrupted by the N-to-Y mutations). Furthermore, we do not fully understand the Reviewer's question(s) here. Certainly we provide evidence in Figures 1 and 2 pointing to a significant role of BAZ1B both in DNA damage recovery and in recruitment of SMARCA5 to sites of damage. Moreover, there is literature precedent for a role of BAZ1B in DNA damage repair (appropriately referenced in the manuscript). This study was originally designed to explore the role of BAZ1A in DNA damage recovery. In the first revision, we had already expanded our characterization of the role of BAZ1B in response to the suggestions of Reviewers #1 and #2.

4. How did BAZ1A (or BAZ1B) recruit SMARCA5 to the chromatin? In other words, what is the molecular mechanism?

It is not currently known precisely how BAZ1A/B-SMARCA5 is recruited to sites of damage.

5. How did Bromodomain-acetyllysine recognition and PHD domain-DNA binding in BAZ1A and BAZ1B interplay with each other in the response of DNA damage?

This question implies that there may be some sort of cooperativity between BAZ1A and BAZ1B. Please see our response to point 2 above. To restate, we do not present evidence suggesting cooperativity or "interplay".

6. The authors did not use other methods to validate BAZ1A bromodomain binding to acetyllsine weakly. This is critical for the whole study since the authors assumed bromodomain-acetyllsine recognition plays an imperative role in BAZ1A modulated DNA damage response.

We disagree strongly with the Reviewer's statement that we did not validate the weak binding of acetyl lysine to BAZ1A bromodomain (BD) in our revised manuscript. Specifically in response to the Reviewer's original query on this aspect, we evaluated the binding of asparagine mutants (N-to-Y) of both BAZ1A (BAZ1A-BD^{NY}) and the BAZ1A gatekeeper mutant (BAZ1A-BD^{EVNY}) to an acetyl lysine peptide (Figure 3f). Because the N-to-Y mutation disrupts the key hydrogen bond between bromodomain and the acetyl group, it is commonly used to probe for acetyl-dependent binding. We see that for both wild-type and gatekeeper-mutant bromodomains, the biolayer interferometry (BLI)

binding signal is strongly reduced, showing that the weak binding is a specific interaction with the bromodomain ligand-binding pocket. In addition, we tested the functional relevance of the very weak acetyl lysine binding to wild-type BAZ1A bromodomain by making an additional cell line expressing the N-to-Y binding site mutant in the background of the *BAZ1A* knockout (Figure 4d). Functional defects were observed for this mutant, suggesting that very weak binding to the BAZ1A ligand pocket is nevertheless important. Given the emphasis the Reviewer placed on this in the first review, we would have expected that the Reviewer would be pleased to see that we had taken it seriously and determined that the weak binding was relevant.

7. Biolayer interferometry is not an ideal method to monitor weak protein-peptide interaction, especially, when K_d is greater than 100M. Authors needed to use other methods to validate K_d value of BAZ1A-BDEV binding to acetylated peptides or histones.

Because of the typo in the comment above, it is difficult to be sure what affinity the Reviewer meant, perhaps 100 μ M? In any case, it is difficult to understand why the Reviewer apparently does not consider BLI (a commonly used method) as reliable. The E-to-V mutant (BAZ1A-BD^{EV}) shows a robust binding signal that fits well to a saturation binding model (Figure 3f). In addition, as described above in response to point 6, we validated the binding as specific through mutagenesis. The test ligand used (a multiply acetylated histone H4 peptide) has no special biological significance, so we do not understand why the exact value of K_D (reported here using BLI) is of sufficient interest to justify substantial additional effort to determine it through a different method.

8. Results from peptide array must be validated by other biochemical (pulldown and western blot) or biophysical methods such as NMR etc.

BLI is a biophysical method, and we used it to validate the interpretation made from the array screen, namely that the BAZ1A-BD^{EV} mutant is better able to bind to acetyl lysine than is wild-type BAZ1A-BD (Figure 3f). The lingering controversy surrounding this conclusion is puzzling. The position of the gatekeeper side chain in peptide complex structures shows that it interacts with the aliphatic portion of the acetyl lysine side chain. It therefore makes structural sense that the gatekeeper is nearly always hydrophobic in functional bromodomains and that a charged residue at this position (as in the case of BAZ1A) would be deleterious to binding.